# Learning Event Completeness for Weakly Supervised Video Anomaly Detection

**Yu Wang**[1]  **Shiwei Chen**[2]

## Abstract

Weakly supervised video anomaly detection (WS-VAD) is tasked with pinpointing temporal intervals containing anomalous events within untrimmed videos, utilizing only video-level annotations. However, a significant challenge arises due to the absence of dense frame-level annotations, often leading to incomplete localization in existing WS-VAD methods. To address this issue, we present a novel LEC-VAD, **L**earning **E**vent **C**ompleteness for Weakly Supervised **V**ideo **A**nomaly **D**etection, which features a dual structure designed to encode both category-aware and category-agnostic semantics between vision and language. Within LEC-VAD, we devise semantic regularities that leverage an anomaly-aware Gaussian mixture to learn precise event boundaries, thereby yielding more complete event instances. Besides, we develop a novel memory bank-based prototype learning mechanism to enrich concise text descriptions associated with anomaly-event categories. This innovation bolsters the text's expressiveness, which is crucial for advancing WS-VAD. Our LEC-VAD demonstrates remarkable advancements over the current state-of-the-art methods on two benchmark datasets XD-Violence and UCF-Crime.

## 1. Introduction

In recent years, video anomaly detection (VAD) has garnered increasing interest due to its widespread applications (Carreira & Zisserman, 2017; Dosovitskiy & et al, 2021; Radford et al., 2021), such as intelligent surveillance (Sultani et al., 2018; Wang et al., 2024), multimedia content review (Zhu et al., 2023), and intelligent manufacturing (Gupta et al., 2024). VAD aims to predict temporal intervals

[1]School of Computer Science and Technology, Tongji University, Shanghai, China. [2]Department of R&D Data, Microsoft Asia-Pacific Technology CO Ltd, Shanghai, China.. Correspondence to: Yu Wang <yuwangtj@yeah.net>.

*Proceedings of the 42 $^{st}$ International Conference on Machine Learning*, Vancouver, Canada. PMLR 267, 2025. Copyright 2025 by the author(s).

of anomaly events in arbitrarily long and untrimmed videos. However, a major challenge faced by VAD is its reliance on costly and dense annotations that precisely mark the start and end of each anomaly event. To mitigate this dependency, weakly supervised VAD (WS-VAD) is introduced. This paradigm facilitates training with only video-level annotations, thereby streamlining the annotation process and enhancing the feasibility of VAD in practical applications.

The majority of contemporary methods (Wu et al., 2024b; Liu et al., 2024; Wu et al., 2024d; Zhou et al., 2023; Cho et al., 2023b; Luo et al., 2021; Ramachandra et al., 2020; Wang et al., 2023a; Feng et al., 2021) for WS-VAD adhere to a systematic pipeline. Initially, the process entails extracting frame-level features by leveraging pre-trained models (Radford et al., 2021; Jia et al., 2021). Following this, these extracted features are input into binary classifiers and integrated with a multiple instance learning (MIL) strategy (Paul et al., 2018). Ultimately, the identification of abnormal events is achieved through the analysis of the predicted anomaly confidences. Despite achieving promising results, such a classification paradigm assigns each video frame to zero or more anomaly categories. During inference, it relies on manually designed post-processing steps to aggregate these frame-level predictions into consecutive anomaly snippets with explicit boundaries. However, this paradigm often results in incomplete and fragmented anomaly segments, as shown in Figure. 1.

To address the challenge of incomplete anomaly event detection in such a paradigm, we propose LEC-VAD, **L**earning **E**vent **C**ompleteness for Weakly Supervised **V**ideo **A**nomaly **D**etection, including two novel mechanisms: memory bank-based prototype learning strategy and Gaussian mixture prior-based local consistency learning. For the former, we resort to textual descriptions of anomaly-category and devise a dual structure to encode both coarse-grained and fine-grained semantics between vision and language. In this procedure, textual expressions of anomaly categories are typically concise and limited in expression, so an innovative memory bank-based prototype learning strategy is developed to strengthen the text's entropy. Besides, we also dynamically enhance text representations by integrating learnable text-condition visual prompts. For the latter, since the model is trained to perform frame-wise prediction, it lacks an explicit understanding of anomaly event bound-

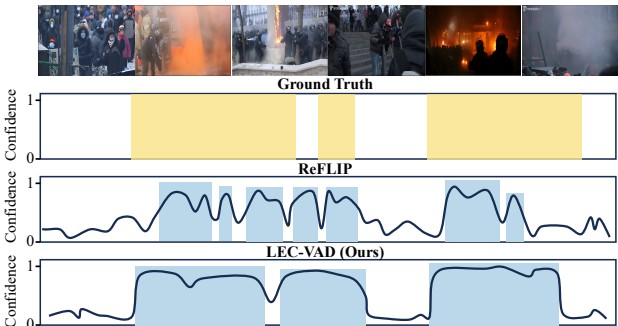

**Figure 1:** Yellow denotes ground truth anomaly intervals, Blue denotes the prediction intervals, and the black lines depict the predicted confidences. We observe that existing methods, exemplified by ReFLIP, typically suffer from the fragmentation of intervals, ultimately yielding incomplete anomaly instances. Instead, our LEC-VAD produces smoother scoring patterns, leading to more complete anomaly intervals.

aries, thus resulting in an objective discrepancy between the classification-based training and localization-based inference. To bridge this gap, we hypothesize that prediction results should exhibit local consistency. To explicitly enforce this prior, we refine anomaly scores with learnable Gaussian mixture masks. This enables anomaly scores to incorporate contextual information from nearby frames, thereby improving the smoothness of predicted anomaly snippets.

The main contributions of this paper are five-fold: 1) We propose LEC-VAD, a novel method to learn event completeness for video anomaly detection with only video-level annotations. 2) A dual structure is devised to mine both category-aware and category-agnostic semantics between vision and language. 3) Since text descriptions of anomaly categories are typically concise and limited in expression, we develop a memory bank-based prototype learning mechanism to enrich textual representations. 4) Based on our hypothesis that prediction results should exhibit local consistency, we propose a prior-driven event completeness learning paradigm to refine anomaly scores equipped with learnable Gaussian mixture masks. 5) Extensive evaluations on the XD-Violence and UCF-Crime datasets have shown that our LEC-VAD achieves state-of-the-art performance. Notably, it demonstrates a significant advantage over existing methods for finer-grained anomaly-event detection, outperforming them by a substantial margin.

## 2. Related Work

### 2.1. Vision-Language Pre-training

Cross-modal vision-language understanding (Jian et al., 2024; Wu et al., 2023; Wei et al., 2024) is a fundamental task that necessitates precise representation alignment between

language and vision. The mainstream approaches can be grouped into two categories, *i.e.,* joint encoder and dual encoder. Methods with a joint encoder (Li et al., 2021; Lu et al., 2019; Zhang et al., 2021) employ a multi-modal encoder to facilitate fine-grained interactions between vision and language. However, despite their promising performance, these methods suffer from a notable drawback in that they necessitate processing each text-image pair individually during inference, leading to significant inefficiencies. In contrast, approaches employing the dual-encoder structure (Radford et al., 2021; Jia et al., 2021) utilize two separate encoders to extract visual and linguistic features. Recently, significant advances have been attributed to large-scale contrastive pre-training within this paradigm, which has dramatically enhanced the performance of numerous multi-modal tasks, including text-image retrieval (Wang et al., 2023b; Jiang & Ye, 2023), visual question answering (Sima et al., 2025; Lin et al., 2023b), and video grounding (Lin et al., 2023a; Xiao et al., 2024). In this paper, we embrace CLIP (Radford et al., 2021) and transfer it for video anomaly detection.

### 2.2. Weakly Supervised Video Anomaly Detection

Weakly Supervised video anomaly detection (WS-VAD) has attracted considerable attention in recent years due to its broad applicability (Wang et al., 2025; 2024; Zhu et al., 2023; Gupta et al., 2024; Sultani et al., 2018) and manageable computational requirements. Sultani *et al.* (Sultani et al., 2018) are pioneers in this field, collecting a large-scale dataset for video anomaly detection annotated at the video level and employing a multiple instance ranking strategy to pinpoint anomalous events. Subsequent research endeavors have focused on various aspects of enhancing WS-VAD performance. One line of research (Zhong et al., 2019; Tian et al., 2021; Zhong et al., 2024; Li et al., 2022; Huang et al., 2022; Zhou et al., 2023) has explored capturing the temporal relationships among video segments. This has been achieved through the use of graph structures (Zhong et al., 2019), self-attention strategies (Tian et al., 2021; Zhou et al., 2023), and transformers (Lv et al., 2023; Huang et al., 2022; Li et al., 2022). These methods provide a deeper understanding of how different parts of a video interact, which is crucial for accurately detecting anomalies. Another paradigm has been to investigate self-training schemes (Yang et al., 2024; Rai et al., 2024; Zhang et al., 2023; Feng et al., 2021). These schemes generate snippet-level pseudo-labels, which are then used to iteratively refine the anomaly scores. This process helps in improving the accuracy of anomaly detection by providing more granular feedback. Besides, Sapkota *et al.* (Sapkota & Yu, 2022) and Zaheer *et al.* (Zaheer et al., 2024) seek solutions to alleviate the impact of false positives. Recently, there has been a surge in efforts to leverage multi-modal knowledge to boost WS-VAD performance. For example, PEL4VAD

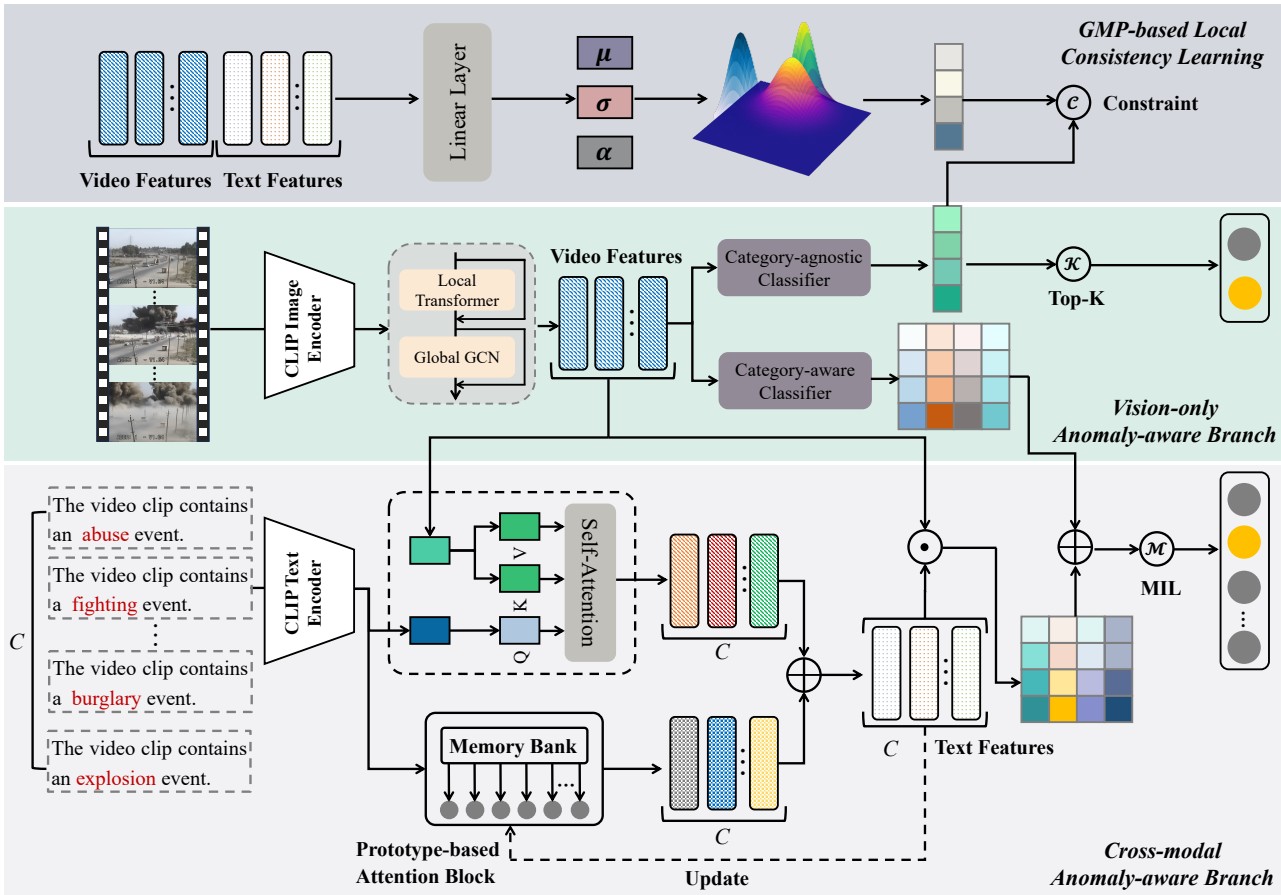

Figure 2: Overview of the proposed LEC-VAD. We develop a dual-structured framework to perceive both category-agnostic and category-aware anomaly clues. We incorporate textual expressions of anomaly categories into WS-VAD, and develop a novel memory bank-based prototype learning mechanism to enrich concise textual descriptions associated with anomaly-event categories. Besides, Gaussian mixture prior-based semantic regularities are devised to learn complete event boundaries.

(Pu et al., 2024), VadCLIP (Wu et al., 2024d), and ITC (Liu et al., 2024) inject text clues of anomaly-event categories for WS-VAD, while PE-MIL (Chen et al., 2024) and MACIL-SD (Yu et al., 2022) introduce audio elements to enhance their capabilities. These innovations have yielded promising results. Wu *et al.* (Wu et al., 2024a) have gone further by developing a framework that can retrieve video anomalies based on language queries or synchronous audio. They (Wu et al., 2024b) have also proposed an open-vocabulary video anomaly detection paradigm to identify both known and novel anomalies in open-world settings. Besides, the use of pre-trained vision-language models, *i.e.,* CLIP (Radford et al., 2021), has also emerged as a powerful tool for extracting robust representations (Zhu & Pang, 2024; Wu et al., 2024d; Dev et al., 2024; Yang et al., 2024). For example, Yang *et al.* (Yang et al., 2024) utilize pre-trained CLIP to generate reliable pseudo-labels. Lv *et al.* (Lv et al., 2023) devise an unbiased multiple-instance strategy to learn invariant representations. STPrompt (Wu et al., 2024c) learns spatio-temporal prompt embeddings based on pre-trained

vision-language models.

Despite significant advances, these existing works fail to harness the principle that predictions should exhibit local consistency. In contrast, our proposed LEC-VAD learns Gaussian mixture masks alongside multi-granularity semantics to ensure the completeness of anomaly event detection.

## 3. Methodology

In this section, we first introduce the proposed dual-structured framework LEC-VAD. Then, we elaborate on the proposed memory bank-based prototype learning mechanism and Gaussian mixture prior-based local consistency learning strategy in Section 3.2 and Section 3.3 respectively.

### 3.1. Overview

**Problem Definition.** Given a dataset consisting of $n$ videos $\mathcal{V} = \{v_i\}_{i=1}^n$ with different annotation granularities, each video $v_i$ is annotated with coarse-grained and fine-grained

video-level labels $y_i \in \{0, 1\}$ and $g_i \in \{0, .., C\}$ respectively, where $C$ is the number of anomaly categories. If $v_i$ is flagged as normal, $y_i = 0$ and $g_i = 0$, otherwise $y_i = 1$ and $g_i$ is assigned a specific category. Coarse-grained WS-VAD needs to predict the likelihood of each snippet containing an anomaly. Fine-grained WS-VAD demands the prediction of specific anomaly event instances, denoted as $\{s_j, e_j, g_j, w_j\}$. Here, $s_j$ and $e_j$ denote the start and end timestamps of the $j$-th anomaly instance, $g_j$ represents the anomaly category, and $w_j$ signifies the prediction confidence for the $j$-th instance.

**Framework.** This paper proposes a novel dual-structured framework, LEC-VAD, for WS-VAD, as illustrated in Figure. 2. The core idea is to learn anomaly event completeness in the absence of dense annotations. To accomplish this goal, we endeavor to capture both category-aware and category-agnostic anomaly knowledge through a dual structure, avoiding subtle clues being overwhelmed by salient features. This allows our model to grasp a comprehensive understanding of event concepts. Besides, to ensure local consistency of predictions, learnable Gaussian mixture masks are developed to improve the smoothness of predicted anomaly snippets. Specifically, we initially employ pre-trained image encoders to extract visual features $X_{video} \in \mathcal{R}^{T \times d}$, where $T$ denotes the number of snippets and $d$ is the feature dimension. As in (Wu et al., 2024d), in order to capture local temporal dependencies, we divide the frame-level features into overlapping windows of equal length. Subsequently, we apply a local transformer layer with constrained attention scopes (Liu et al., 2021), ensuring no information exchange between these windows. This process results in the enhanced feature $X_l \in \mathcal{R}^{T \times d}$. Then, to further capture global temporal dependencies, we apply a GCN module on $X_l$. Specifically, following the setup in (Wu et al., 2024d), global temporal associations of features are modeled based on their similarity. This process can be summarized as follows:

$$F_{video} = GELU(Softmax(A)X_l W),$$
$$A = \frac{X_l X_l^T}{||X_l||_2 \cdot ||X_l||_2}, \tag{1}$$

where $A$ is the adjacency matrix, the Softmax operation is used to ensure that the sum of each row equals one, $W$ is the learnable weight, and $F_{video}$ is the enhanced video feature.

Besides, a pre-trained text encoder is adopted to extract linguistic features of all anomaly-event categories $F_{text} \in \mathcal{R}^{(C+1) \times d}$. Then, we apply a fully-connected layer, followed by a softmax operation, on $F_{video}$ to perform binary classification for category-agnostic anomaly detection and multivariate classification for category-aware anomaly detection, yielding prediction results $s^b \in \mathcal{R}^{T \times 2}$ and $s^m \in \mathcal{R}^{T \times C}$. For $s^b$, we compute the average of the top-$K$ scores over the temporal dimension, specifically for the $c$-th

category (belongs to $\{0, 1\}$) category, as follows:

$$p_c^b = \max_{\substack{H \subseteq \{1, ..., T\} \\ |H| = K}} \frac{1}{K} \sum_{j \in H} s_{jc}^b, \tag{2}$$

where $H$ denotes the index set of $K$ snippets, and $p_c^b$ is the category-agnostic predictions. Then, a binary cross-entropy loss is imposed on $p^b$ and ground truth $y$:

$$\mathcal{L}_{agnostic} = - \sum_{c \in \{0,1\}} y_c \log(p_c^b). \tag{3}$$

Furthermore, although $s^m$ reflects category-aware anomaly information, it lacks associations with textual descriptions. To this end, we model cross-modal interactions through a cross-attention operation. Here the text feature $F_{text}$ serves as the query, while the video feature $F_{video}$ acts as the key and value, resulting in an enhanced representation $F_{tv} \in \mathcal{R}^{(C+1) \times d}$. The computation is outlined as follows:

$$Q = F_{text} \cdot W_q, K = F_{video} \cdot W_k, V = F_{video} \cdot W_v,$$
$$F_{tv} = softmax(\frac{Q \cdot K^T}{\sqrt{d}}) \cdot V, \tag{4}$$

where $W_q$, $W_k$, and $W_v$ are learnable projection matrices in $\mathcal{R}^{d \times d}$. Furthermore, given the concise and limited expressive nature of textual descriptions for anomaly-event categories, we develop a memory bank-based mechanism to store category prototypes. These prototypes are leveraged to acquire enhanced text descriptions $F_{aug}$. The ultimate text representation, $F_{ctg}$, is a fusion of $F_{tv}$ and $F_{aug}$, formulated as $F_{ctg} = F_{tv} + F_{aug}$. Furthermore, we utilize an external memory bank to strengthen text expressiveness, acquiring an enhanced feature $F_{aug}$. After that, we perform category-aware detection through a dot product of $F_{ctg}$ and video features $F_{video}$, which is presented as follows:

$$s^{tv} = norm(F_{video}) \cdot norm(F_{ctg}^T), \tag{5}$$

where $norm$ denotes the $\ell_2$ normalization. Finally, we obtain the category-aware anomaly detection logit by integrating $s^{tv}$ and $s^m$, formulated as $s^{aware} = s^m + s^{tv}$. This logit is then utilized via the MIL principle to yield video-level classification outcomes. More precisely, for the $c$-th anomaly category, the logit $p_c^m$ is calculated as follows:

$$p_c^m = \max_{\substack{H \subseteq \{1, ..., T\} \\ |H| = K}} \frac{1}{K} \sum_{j \in H} s_{jc}^{aware}. \tag{6}$$

Subsequently, a cross-entropy constraint is imposed to supervise the process of fine-grained anomaly classification:

$$\mathcal{L}_{aware} = - \sum_{c=0}^{C} g_c \log(p_c^m). \tag{7}$$

To learn complete anomaly events and improve the smoothness of snippets, we hypothesize that prediction scores should be locally consistent. Based on this hypothesis, we propose a Gaussian mixture prior-based local consistency learning mechanism, encoding multiple anomaly categories as GMM components, thereby generating anomaly constraint scores $s_{gmm} \in \mathcal{R}^T$. $s_{gmm}$ is then adopted to regularize the predicted anomaly scores $s^b$ as follows:

$$\mathcal{L}_{gmm} = \sqrt{\sum_{t=1}^{T}(s_{gmm}[t] - s^b[t,1])^2}. \tag{8}$$

Besides, to guarantee the consistency of the category-agnostic anomaly score, i.e., $1 - s^m[t,0]$, and category-aware anomaly score, i.e., $s^b[t,1]$, for each snippet, we introduce a $\ell_1$ regularization loss, formulated as follows:

$$\mathcal{L}_{reg} = \frac{1}{T}\sum_{t=1}^{T}|1 - s^m[t,0] - s^b[t,1]|. \tag{9}$$

Overall, the optimization loss of the proposed LEC-VAD is summarized as follows:

$$\mathcal{L}_{all} = \mathcal{L}_{agnostic} + \mathcal{L}_{aware} + \lambda\mathcal{L}_{gmm} + \gamma\mathcal{L}_{reg}, \tag{10}$$

where $\lambda$ and $\gamma$ are weights that balance different terms.

### 3.2. Memory Bank-based Prototype Learning

Since text descriptions of anomaly-event categories are generally concise and expressive-limited, we construct an external memory bank to retain a group of textual prototypes, denoted as $\mathcal{M} \in \mathcal{R}^{(C+1)\times d}$. These prototypes are subsequently leveraged to strengthen textual representations. In particular, we initialize $\mathcal{M}$ with the original CLIP features corresponding to the anomaly-event categories. During training, the enhanced text features $F_{tv}$ are concatenated with the contents of $\mathcal{M}$ to produce context-aware features $F_{cte} \in \mathcal{R}^{(2C+2)\times d}$. Subsequently, we apply $m$ self-attention blocks on $F_{cte}$ to absorb semantic knowledge from prototype features $\mathcal{M}$, thereby acquiring the augmented representation $F'_{cte} \in \mathcal{R}^{(2C+2)\times d}$. For $F'_{cte}$, we extract the first $C + 1$ representations along the first dimension, denoted as $F_{aug} \in \mathcal{R}^{(T+1)\times d}$, as the refined text representations. Meanwhile, prototype features in $\mathcal{M}$ are updated in a momentum-based fashion, specifically as follows:

$$\mathcal{M} = \eta \times \mathcal{M} + (1 - \eta) \times F_{aug}, \tag{11}$$

where $\eta$ denotes the momentum coefficient for an update.

### 3.3. Gaussian Mixture Prior-based (GMP-based) Local Consistency Learning

This paper proposes the hypothesis that prediction scores ought to exhibit local consistency. To explicitly enforce this prior and learn complete instances, we model anomaly scores with learnable Gaussian Mixture Models (GMMs) for each temporal position $t$, where each component of GMM is tailored to encode category-specific anomaly mask. In particular, we first concatenate $F_{aug}$ with the visual feature of each snippet $F_{video}$ by a broadcast mechanism in Python to get integrated multi-modal representation $F_m \in \mathcal{R}^{T\times C\times d}$. Based on $F_m$, we utilize a shared fully-connected layer to predict Gaussian kernels $\{\sigma_c^t, \mu_c^t\}_{t=1}^{T}$ of Gaussian masks for each category. Finally, we generate anomaly scores $s_{gauss}(t)$ at $t$-th temporal position from the $t$-th Gaussian mixture mask. The detailed procedure is as follows:

$$
\begin{aligned}
G^t &= \sum_{c=0}^{C} \alpha_c^t G_c^t(t), \ where \ \alpha_c^t = s^m[t,c], \\
G_c^t &= exp(-\frac{\beta(j/T - \mu_c^t)^2}{(\sigma_c^t)^2})_{j=1}^{T}, \\
s_{gmm}(t) &= \{G^t(t)\}_{t=1}^{T},
\end{aligned} \tag{12}
$$

where $\alpha_c^t$ indicates the probability of each abnormal event occurring, assigned $s^m[t,c]$. $\beta$ controls the variance of the Gaussian mask $G_c^t$. In this manner, these masks exhibit local smoothness, rendering them suitable for constraining anomaly scores $s^b$, as defined in Eq. 8.

### 3.4. Inference

During inference, for the coarse-grained WS-VAD, we calculate the average value of $1 - s^m[t,0]$ and $s^b[t,1]$ as the $t$-th frame's anomaly confidence. For fine-grained WS-VAD, we adopt a two-step thresholding strategy to generate anomaly instances. Specifically, we first retain these anomaly categories with video-level activations exceeding a predefined threshold $r_{cls}$. Then, for each retained anomaly category, we select snippets with fine-grained matching scores in $s^m$ exceeding the threshold $r_{ano}$ as candidates. These temporally consecutive candidates are merged to form anomaly instances. Following AutoLoc (Shou et al., 2018), the outer-inner score of each instance based on $s^m$ is regarded as the confidence score of each instance. Based on these confidence scores, we apply a non-maximal suppression (NMS) to avoid redundant proposals.

## 4. Experiments

### 4.1. Experimental Settings

**Datasets. UCF-Crime** (Sultani et al., 2018) consists of 1900 untrimmed videos, spanning across 13 categories of anomalous events. These videos are collected from surveil-

| Supervision | Modality | Methods | Feature | AP(%) |
|---|---|---|---|---|
| Unsupervised | RGB+Audio | LTR (Hasan et al., 2016) | I3D+VGGish | 30.77 |
| Weakly Supervised | RGB+Audio | CTRFD (Wu & Liu, 2021) | I3D+VGGish | 75.90 |
| | | WS-AVVD (Wu et al., 2022) | I3D+VGGish | 78.64 |
| | | ECU (Zhang et al., 2023) | I3D+VGGish | 81.43 |
| | | DMU (Zhou et al., 2023) | I3D+VGGish | 81.77 |
| | | MACIL-SD (Yu et al., 2022) | I3D+VGGish | 83.40 |
| | | PE-MIL (Chen et al., 2024) | I3D+VGGish | 88.21 |
| | RGB | RAD (Sultani et al., 2018) | C3D | 73.20 |
| | | RTFML (Tian et al., 2021) | I3D | 77.81 |
| | | ST-MSL (Li et al., 2022) | I3D | 78.28 |
| | | LA-Net (Pu & Wu, 2022) | I3D | 80.72 |
| | | CoMo (Cho et al., 2023a) | I3D | 81.30 |
| | | DMU (Zhou et al., 2023) | I3D | 82.41 |
| | | PEL4VAD (Pu et al., 2024) | I3D | 85.59 |
| | | PE-MIL (Chen et al., 2024) | I3D | 88.05 |
| | | LEC-VAD (Ours) | I3D | **88.47** |
| | | OVVAD (Wu et al., 2024b) | CLIP | 66.53 |
| | | CLIP-TSA (Joo et al., 2023) | CLIP | 82.19 |
| | | IFS-VAD (Zhong et al., 2024) | CLIP | 83.14 |
| | | TPWNG (Yang et al., 2024) | CLIP | 83.68 |
| | | VadCLIP (Wu et al., 2024d) | CLIP | 84.51 |
| | | ITC (Liu et al., 2024) | CLIP | 85.45 |
| | | ReFLIP (Dev et al., 2024) | CLIP | 85.81 |
| | | LEC-VAD (Ours) | CLIP | **86.56** |

Table 1: Coarse-grained comparisons on XD-Violence.

| Supervision | Methods | Feature | AUC(%) |
|---|---|---|---|
| Unsupervised | LTR (Hasan et al., 2016) | I3D+VGGish | 50.60 |
| | GODS (Wang & Cherian, 2019) | I3D | 70.46 |
| | GCL (Zaheer et al., 2022) | I3D | 71.04 |
| Weakly Supervised | TCN-CIBL (Zhang et al., 2019) | C3D | 78.66 |
| | GCN-Anomaly (Zhong et al., 2019) | C3D | 81.08 |
| | GLAWS (Zaheer et al., 2020) | C3D | 83.03 |
| | LEC-VAD (Ours) | C3D | **84.75** |
| | RAD (Sultani et al., 2018) | I3D | 77.92 |
| | WS-AVVD (Wu et al., 2022) | I3D | 82.44 |
| | RTFML (Tian et al., 2021) | I3D | 84.30 |
| | CTRFD (Wu & Liu, 2021) | I3D | 84.89 |
| | LA-Net (Pu & Wu, 2022) | I3D | 85.12 |
| | ST-MSL (Li et al., 2022) | I3D | 85.30 |
| | IFS-VAD (Zhong et al., 2024) | I3D | 85.47 |
| | NG-MIL (Park et al., 2023) | I3D | 85.63 |
| | CLAV (Cho et al., 2023b) | I3D | 86.10 |
| | ECU (Zhang et al., 2023) | I3D | 86.22 |
| | DMU (Zhou et al., 2023) | I3D | 86.75 |
| | PE-MIL (Chen et al., 2024) | I3D | 86.83 |
| | MGFN (Chen et al., 2023) | I3D | 86.98 |
| | LEC-VAD (Ours) | I3D | **88.21** |
| | Ju *et al.* (Ju et al., 2022) | CLIP | 84.72 |
| | OVVAD (Wu et al., 2024b) | CLIP | 86.40 |
| | IFS-VAD (Zhong et al., 2024) | CLIP | 86.57 |
| | UMIL (Lv et al., 2023) | CLIP | 86.75 |
| | CLIP-TSA (Joo et al., 2023) | CLIP | 87.58 |
| | TPWNG (Yang et al., 2024) | CLIP | 87.79 |
| | VadCLIP (Wu et al., 2024d) | CLIP | 88.02 |
| | STPrompt (Wu et al., 2024c) | CLIP | 88.08 |
| | ReFLIP (Dev et al., 2024) | CLIP | 88.57 |
| | ITC (Liu et al., 2024) | CLIP | 89.04 |
| | LEC-VAD (Ours) | CLIP | **89.97** |

Table 2: Coarse-grained comparisons on UCF-Crime.

| Methods | mAP@IoU | | | | | |
|---|---|---|---|---|---|---|
| | 0.1 | 0.2 | 0.3 | 0.4 | 0.5 | AVG |
| RAD (Sultani et al., 2018) | 22.72 | 15.57 | 9.98 | 6.20 | 3.78 | 11.65 |
| AVVD (Wu et al., 2022) | 30.51 | 25.75 | 20.18 | 14.83 | 9.79 | 20.21 |
| VadCLIP (Wu et al., 2024d) | 37.03 | 30.84 | 23.38 | 17.90 | 14.31 | 24.70 |
| ITC (Liu et al., 2024) | 40.83 | 32.80 | 25.42 | 19.65 | 15.47 | 26.83 |
| ReFLIP (Dev et al., 2024) | 39.24 | 33.45 | 27.71 | 20.86 | 17.22 | 27.36 |
| LEC-VAD (Ours) | **48.78** | **40.94** | **34.28** | **28.02** | **23.68** | **35.14** |

Table 3: Fine-grained comparisons on XD-Violence.

| Methods | mAP@IoU | | | | | |
|---|---|---|---|---|---|---|
| | 0.1 | 0.2 | 0.3 | 0.4 | 0.5 | AVG |
| RAD (Sultani et al., 2018) | 5.73 | 4.41 | 2.69 | 1.93 | 1.44 | 3.24 |
| AVVD (Wu et al., 2022) | 10.27 | 7.01 | 6.25 | 3.42 | 3.29 | 6.05 |
| VadCLIP (Wu et al., 2024d) | 11.72 | 7.83 | 6.40 | 4.53 | 2.93 | 6.68 |
| ITC (Liu et al., 2024) | 13.54 | 9.24 | 7.45 | 5.46 | 3.79 | 7.90 |
| ReFLIP (Dev et al., 2024) | 14.23 | 10.34 | 9.32 | 7.54 | 6.81 | 9.62 |
| LEC-VAD (Ours) | **19.65** | **17.17** | **14.37** | **9.45** | **7.18** | **13.56** |

Table 4: Fine-grained comparisons on UCF-Crime.

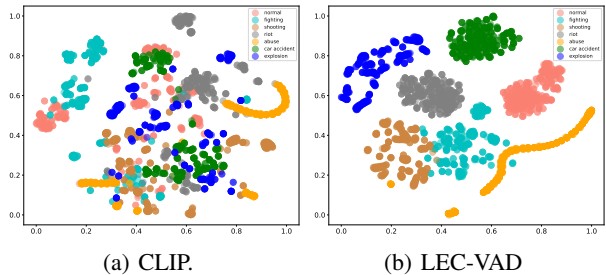

(a) CLIP.      (b) LEC-VAD

Figure 3: The representation visualization for 7 categories in XD-Violence testing set by using the t-SNE method (Van der Maaten & Hinton, 2008).

lance footage captured in various indoor settings and street scenarios. We adhere to a standard data split, where the training set and testing set comprise 1610 and 290 videos, respectively. **XD-Violence** (Wu et al., 2020) is a larger-scale benchmark comprising 4754 untrimmed videos sourced from movies and YouTube, encompassing 6 violence event categories. 3954 videos and 800 videos are employed for training and testing respectively.

**Evaluation Protocol.** For coarse-grained WS-VAD, we adopt frame-level Average Prevision (AP) for XD-Violence and frame-level AUC for UCF-Crime. For fine-grained WS-VAD, we follow previous works (Wu et al., 2024d; Liu et al., 2024) and compute the mean Average Precision (mAP) across IoU thresholds from 0.1 to 0.5 in increments of 0.1. Besides, an average of mAP (AVG) is also reported for a more comprehensive evaluation.

**Implementation Details.** The pre-trained text encoder of CLIP (ViT-B/16) is adopted and multiple vision encoders including I3D (Carreira & Zisserman, 2017), C3D (Tran et al., 2015), and the CLIP (ViT-B/16) are explored to extract frame features. The value of $K$ is determined as $K = max(\lfloor T/16 \rfloor, 1)$, and the momentum coefficient $\eta$ is set to 0.99. We adopt the AdamW optimizer and train our LEC-VAD with a batch size of 64. The learning rate is set to $3e\text{-}5$ and the model is trained for 10 epochs. We apply NMS with an IoU threshold of 0.5, and set the threshold $r_{cls}$, and $r_{ano}$ to 0.1 and 0.2. The hyper-parameters $\beta$, $\lambda$, $\gamma$, and $m$ are explored in the experimental sections (Figure 5).

| VOB | CMB | mAP@IoU | | | | | |
|---|---|---|---|---|---|---|---|
| | | 0.1 | 0.2 | 0.3 | 0.4 | 0.5 | AVG |
| ✓ | | 42.86 | 33.82 | 30.58 | 24.66 | 18.92 | 30.17 |
| | ✓ | 46.03 | 38.87 | 32.18 | 27.02 | 22.04 | 33.23 |
| ✓ | ✓ | **48.78** | **40.94** | **34.28** | **28.02** | **22.68** | **34.94** |

Table 5: Explorations of the model structure. "VOB" denotes the vision-only anomaly-aware branch, and "CMB" denotes the cross-modal anomaly-aware branch.

| $\mathcal{L}_{reg}$ | $\mathcal{L}_{gmm}$ | mAP@IoU | | | | | |
|---|---|---|---|---|---|---|---|
| | | 0.1 | 0.2 | 0.3 | 0.4 | 0.5 | AVG |
| | | 44.59 | 36.34 | 28.97 | 22.83 | 17.45 | 30.04 |
| ✓ | | 47.04 | 38.52 | 31.23 | 25.38 | 20.26 | 32.49 |
| | ✓ | 46.69 | 39.51 | 32.17 | 27.16 | 22.01 | 33.51 |
| ✓ | ✓ | **48.78** | **40.94** | **34.28** | **28.02** | **22.68** | **34.94** |

Table 6: Ablation studies of loss function.

| VAP | PMB | mAP@IoU | | | | | |
|---|---|---|---|---|---|---|---|
| | | 0.1 | 0.2 | 0.3 | 0.4 | 0.5 | AVG |
| | | 42.17 | 34.66 | 27.98 | 22.47 | 17.24 | 28.90 |
| ✓ | | 46.24 | 38.75 | 31.97 | 26.63 | 20.05 | 32.73 |
| | ✓ | 46.01 | 38.24 | 31.80 | 26.56 | 20.04 | 32.53 |
| ✓ | ✓ | **48.78** | **40.94** | **34.28** | **28.02** | **22.68** | **34.94** |

Table 7: Ablation studies of text enhancement. "VAP" denotes the vision-aware prompt strategy and "PMB" denotes the prototype-based memory bank mechanism.

## 4.2. Main Results

We undertake a comprehensive evaluation of LEC-VAD's anomaly detection capabilities by benchmarking it against widely-used approaches across two datasets. The resultant findings underscore that our LEC-VAD achieves state-of-the-art performance across all evaluation metrics and at various granularity levels. We will elaborate on it below.

First, we assess coarse-grained anomaly detection on XD-Violence and UCF-Crime, with the respective results presented in Table 1 and Table 2. For XD-Violence, we employ I3D and CLIP features of videos, achieving state-of-the-art performance. Specifically, our method achieves an impressive 88.47 AP using I3D features, marking a 0.42 absolute gain compared to PE-MIL. Similarly, utilizing CLIP features results in an 86.56 AP, which acquires a 0.75 absolute gain over ReFLIP. Notably, our approach even surpasses methods that incorporate additional audio features. For UCF-Crime, we adopt C3D, I3D, and CLIP features of videos, all achieving remarkable results. Specifically, our method achieves an impressive 84.75 AP when employing C3D features, making a significant 1.72 absolute improvement compared to CLAWS. Similarly, our method demonstrated an 88.21 AP with I3D features and an even higher 89.97 AP with CLIP features, translating to 1.23 and 0.93 absolute gains respectively, compared to the best practices. These presented results demonstrate our model's superiority and robustness when utilizing diverse visual features.

Furthermore, our method manifests more powerful advantages for fine-grained anomaly detection. As shown in Table 3, our method exhibits significant improvements across all evaluation metrics on the XD-Violence, culminating in an outstanding AVG of 35.14. This represents a considerable improvement of 28.44% over the ReFLIP-VAD method. Also, on the more challenging UCF-Crime, as shown in Ta-

ble 4, our method consistently exhibits remarkable improvements across all evaluation metrics, achieving an amazing AVG of 13.56. This represents a substantial improvement of 40.96% over the ReFLIP-VAD method. Notably, our LEC-VAD brings more gains when using more stringent evaluation criteria (larger IoU). For instance, on XD-Violence, LEC-VAD gets a 37.51% relative boost when IoU is 0.5 and a 19.47% relative boost when IoU is 0.1. This reveals that LEC-VAD can learn more complete instances. A similar phenomenon occurs in UCF-Crime. These observations reveal the superiority of our proposed algorithm in discerning subtle distinctions among diverse anomaly events.

To further investigate our model's capability in learning discriminative visual representations for different anomaly categories, we visualize the feature distributions extracted by the original CLIP and our model on the XD-Violence test set, as shown in Figure 3. From the visualization, we observe that the distribution of original CLIP features exhibits significant overlap and confusion among different categories. In contrast, our model's extracted features for each category form more concentrated clusters with well-defined boundaries. This reveals that our model possesses the capability to learn discriminative features despite the absence of explicit snippet-level supervision. This property is advantageous for our LEC-VAD, enabling it to detect anomaly events with greater completeness and confidence.

## 4.3. Ablation Study

We conduct thorough ablation studies on XD-Violence and systematically report fine-grained anomaly detection results across different evaluation metrics to dissect the contributions of diverse factors to the overall performance.

**Model components.** In Table 5, we conduct an in-depth analysis to examine the impact of various model components on anomaly detection performance. Specifically, we investigate the utilities of the vision-only anomaly-aware branch (VOB) and the cross-modal anomaly-aware branch (CMB), aiming to gain insights into how these components contribute to the overall performance of our model. Overall, the fully intact model demonstrates superiority over its castrated counterparts across all evaluated metrics. This ob-

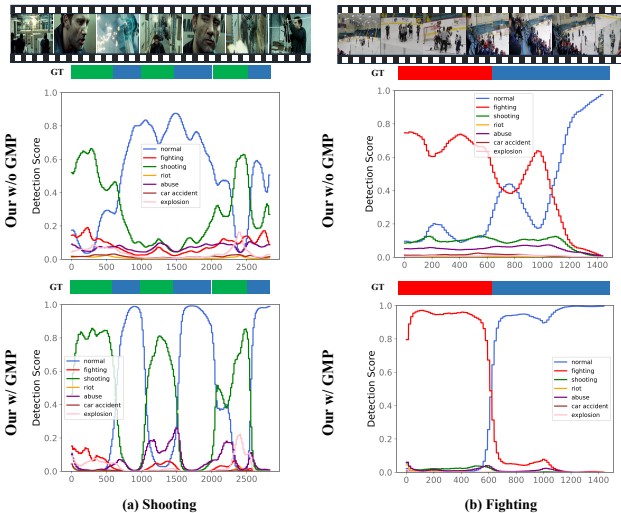

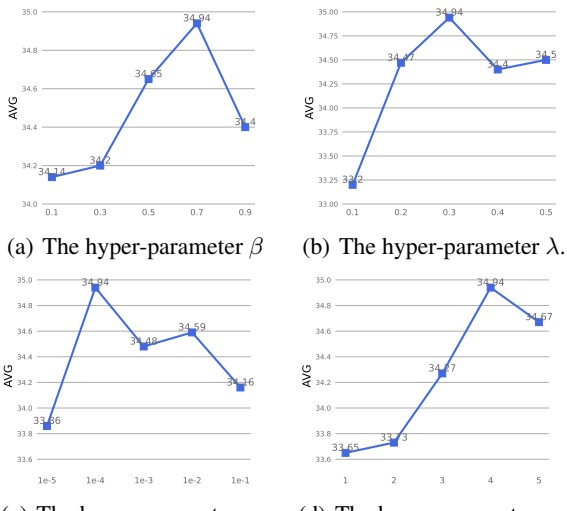

(a) Shooting  (b) Fighting

Figure 4: Visualization about the utility of our GMP.

(a) The hyper-parameter $\beta$  (b) The hyper-parameter $\lambda$.

(c) The hyper-parameter $\gamma$.  (d) The hyper-parameter $m$

Figure 5: Effects of hyper-parameters.

servation indicates that both VOB and CMB are beneficial to WS-VAD. Notably, CMB brings more gains compared to VOB, with an AVG of 33.23 versus 30.17. This reflects the importance of cross-modal interactions for WS-VAD.

**Loss functions.** In Table 6, we delve into the impact of Gaussian mixture prior and score regularization by carefully controlling the loss terms $\mathcal{L}_{reg}$ and $\mathcal{L}_{gmm}$ within the loss function. Undoubtedly, the overall loss achieves optimal performance compared to the castrated counterparts. Besides, we observe that as we apply more rigorous constraint evaluation criteria (characterized by larger IoU values), using $\mathcal{L}_{gmm}$ can bring more gains. This phenomenon suggests that introducing the Gaussian mixture prior can help detect more complete abnormal events. We also provide some visualized comparisons in Figure 4, which illustrates the

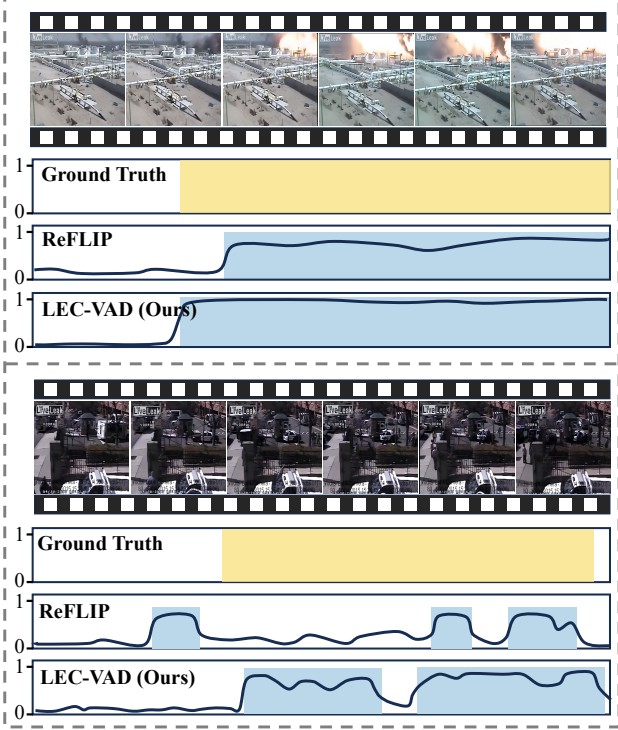

Figure 6: Visualization of coarse-grained anomaly detection results compared with ReFLIP.

differences in outcomes when incorporating Gaussian-based scores versus not using them. From the results, we observe that employing GMP yields more comprehensive and sound outcomes.

**Text enhancement.** This paper introduces textual descriptions of anomaly categories to facilitate cross-modal semantic interactions. Consequently, we explore the proposed enhancement mechanism for concise text descriptions, as illustrated in Table 7. We observe that employing the developed vision-aware prompt strategy (VAP) and prototype-based memory bank mechanism significantly improves anomaly detection performance compared to the variants lacking these components. The integration of these two mechanisms further obtains an impressive AVG of 34.94, achieving an improvement of 20.90% over the variant without them.

**Hyper-parameters.** We conduct an extensive exploration of the hyper-parameter configuration for $\beta$, $\lambda$, $\gamma$, and $m$, as illustrated in Figure 5. For $\beta$, we incrementally adjust it from 0.1 to 0.9 in increments of 0.2, and the optimal is obtained at $\beta = 0.7$, demonstrating that relatively larger variances of Gaussian masks are beneficial to guarantee local consistency. The hyper-parameters $\lambda$ and $\gamma$ are meticulously tuned to ascertain their optimal values, ultimately determining $\lambda = 0.3$ and $\gamma = 1e\text{-}4$ as the most effective configuration. Besides, our investigation into the number of

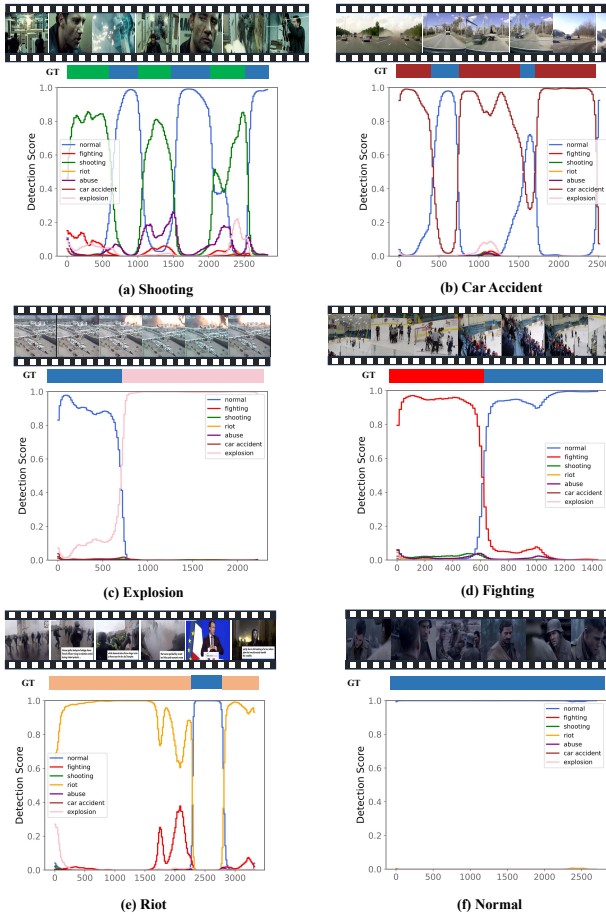

Figure 7: Visualization of fine-grained predictions.

anomaly event displays coarse-grained detection results at the bottom. We observe that ReFLIP incorrectly identifies segments featuring people (such as pedestrians) as clues to anomaly events in this example, failing to discern behavioral differences. In contrast, our LEC-VAD accurately focuses on the behavioral characteristics associated with an "arrest", demonstrating its sensitivity to specific contextual and behavioral clues.

Furthermore, Figure 7 presents some fine-grained visualization results to investigate the inter-relationships between predictive scores across multiple anomaly classes. Instead of a blended outcome from multiple categories, we observed that the true anomaly class has been accurately identified.

## 5. Conclusion

This paper proposed a novel LEC-VAD for WS-VAD, engineered to encode category-aware and category-agnostic semantics of anomaly events. We hypothesize local consistency in predictions and develop a prior-driven learning mechanism with learnable Gaussian masks. Besides, a memory bank-based prototype learning mechanism was proposed to enrich textual features. Overall, LEC-VAD achieved remarkable advances in XD-Violence and UCF-Crime.

## Impact Statement

This paper presents work whose goal is to advance the field of Machine Learning. There are many potential societal consequences of our work, none which we feel must be specifically highlighted here.

## Acknowledgements

This work was supported in part by the National Natural Science Foundation of China under Grant 62406226, in part sponsored by Shanghai Sailing Program under Grant 24YF2748700, and in part sponsored by Tongji University Independent Original Cultivation Project under Grant 22120240326.

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

prototype-based attention blocks $m$, revealed that $m = 4$ is sufficient for achieving optimal performance. Increasing $m$ is observed to potentially induce overfitting.

### 4.4. Qualitative Results

To further demonstrate the superiority of the proposed LEC-VAD for WS-VAD, we have visualized several examples in Figure 6 and Figure 7. Figure 6 compares the anomaly detection outcomes of our LEC-VAD with the state-of-the-art ReFLIP method. In the upper section of the figure, an example depicting the "explosion" anomaly event showcases the coarse-grained detection results of both methods. Upon examination, it becomes evident that our LEC-VAD is capable of detecting more complete anomaly events with greater confidence than the highly competitive ReFLIP method. It appears that ReFLIP primarily relies on the visual presence of fire to judge explosion anomalies in this scenario, whereas our method possesses a deeper understanding of the semantic content and context associated with explosions. Consequently, our LEC-VAD can detect more complete instances. An additional example depicting the "arrest"

Chen, Y., Liu, Z., Zhang, B., Fok, W., Qi, X., and Wu, Y.-C. Mgfn: Magnitude-contrastive glance-and-focus network for weakly-supervised video anomaly detection. In *Proceedings of the AAAI conference on artificial intelligence*, pp. 387–395, 2023.

Cho, M., Kim, M., Hwang, S., Park, C., Lee, K., and Lee, S. Look around for anomalies: Weakly-supervised anomaly detection via context-motion relational learning. In *Proceedings of the IEEE Conference on Computer Vision and Pattern Recognition*, pp. 12137–12146, 2023a.

Cho, M., Kim, M., Hwang, S., Park, C., Lee, K., and Lee, S. Look around for anomalies: Weakly-supervised anomaly detection via context-motion relational learning. In *Proceedings of the IEEE Conference on Computer Vision and Pattern Recognition*, pp. 12137–12146, 2023b.

Dev, P. P., Hazari, R., and Das, P. Reflip-vad: Towards weakly supervised video anomaly detection via vision-language model. *IEEE Transactions on Circuits and Systems for Video Technology*, 2024.

Dosovitskiy, A. and et al. An image is worth 16x16 words: Transformers for image recognition at scale. In *Proceedings of the International Conference on Learning Representations*, 2021.

Feng, J.-C., Hong, F.-T., and Zheng, W.-S. Mist: Multiple instance self-training framework for video anomaly detection. In *Proceedings of the IEEE Conference on Computer Vision and Pattern Recognition*, pp. 14009–14018, 2021.

Gupta, H., Sharma, S., and Agrawal, S. Artificial intelligence-based anomalies detection scheme for identifying cyber threat on iot-based transport network. *IEEE Transactions on Consumer Electronics*, 70(1):1716–1724, 2024.

Hasan, M., Choi, J., Neumann, J., Roy-Chowdhury, A. K., and Davis, L. S. Learning temporal regularity in video sequences. In *Proceedings of the IEEE Conference on Computer Vision and Pattern Recognition*, pp. 733–742, 2016.

Huang, C., Liu, C., Wen, J., Wu, L., Xu, Y., Jiang, Q., and Wang, Y. Weakly supervised video anomaly detection via self-guided temporal discriminative transformer. *IEEE Transactions on Cybernetics*, 54(5):3197–3210, 2022.

Jia, C., Yang, Y., Xia, Y., Chen, Y.-T., Parekh, Z., Pham, H., Le, Q., Sung, Y.-H., Li, Z., and Duerig, T. Scaling up visual and vision-language representation learning with noisy text supervision. In *Proceedings of the International Conference on Machine Learning*, pp. 4904–4916, 2021.

Jian, Y., Gao, C., and Vosoughi, S. Bootstrapping vision-language learning with decoupled language pre-training. In *Proceedings of the Neural Information Processing Systems*, pp. 57–72, 2024.

Jiang, D. and Ye, M. Cross-modal implicit relation reasoning and aligning for text-to-image person retrieval. In *Proceedings of the IEEE Conference on Computer Vision and Pattern Recognition*, pp. 2787–2797, 2023.

Joo, H. K., Vo, K., Yamazaki, K., and Le, N. Clip-tsa: Clip-assisted temporal self-attention for weakly-supervised video anomaly detection. In *Proceedings of the IEEE International Conference on Image Processing*, pp. 3230–3234, 2023.

Ju, C., Han, T., Zheng, K., Zhang, Y., and Xie, W. Prompting visual-language models for efficient video understanding. In *Proceedings of the European Conference on Computer Vision*, pp. 105–124, 2022.

Li, J., Selvaraju, R., Gotmare, A., Joty, S., Xiong, C., and Hoi, S. C. H. Align before fuse: Vision and language representation learning with momentum distillation. In *Proceedings of the Neural Information Processing Systems*, pp. 9694–9705, 2021.

Li, S., Liu, F., and Jiao, L. Self-training multi-sequence learning with transformer for weakly supervised video anomaly detection. In *Proceedings of the AAAI Conference on Artificial Intelligence*, pp. 1395–1403, 2022.

Lin, K. Q., Zhang, P., Chen, J., Pramanick, S., Gao, D., Wang, A. J., Yan, R., and Shou, M. Z. Univtg: Towards unified video-language temporal grounding. In *Proceedings of the IEEE International Conference on Computer Vision*, pp. 2794–2804, 2023a.

Lin, W., Chen, J., Mei, J., Coca, A., and Byrne, B. Fine-grained late-interaction multi-modal retrieval for retrieval augmented visual question answering. In *Proceedings of the Neural Information Processing Systems*, pp. 22820–22840, 2023b.

Liu, T., Lam, K.-M., and Bao, B.-K. Injecting text clues for improving anomalous event detection from weakly labeled videos. *IEEE Transactions on Image Processing*, 33:5907–5920, 2024.

Liu, Z., Lin, Y., Cao, Y., Hu, H., Wei, Y., Zhang, Z., Lin, S., and Guo, B. Swin transformer: Hierarchical vision transformer using shifted windows. In *Proceedings of the IEEE International Conference on Computer Vision*, pp. 10012–10022, 2021.

Lu, J., Batra, D., Parikh, D., and Lee, S. Vilbert: Pre-training task-agnostic visiolinguistic representations for vision-and-language tasks. In *Proceedings of the Neural Information Processing Systems*, pp. 13–23, 2019.

Luo, W., Liu, W., Lian, D., and Gao, S. Future frame prediction network for video anomaly detection. *IEEE Transactions on Pattern Analysis and Machine Intelligence*, 44 (11):7505–7520, 2021.

Lv, H., Yue, Z., Sun, Q., Luo, B., Cui, Z., and Zhang, H. Unbiased multiple instance learning for weakly supervised video anomaly detection. In *Proceedings of the IEEE Conference on Computer Vision and Pattern Recognition*, pp. 8022–8031, 2023.

Park, S., Kim, H., Kim, M., Kim, D., and Sohn, K. Normality guided multiple instance learning for weakly supervised video anomaly detection. In *Proceedings of the IEEE Winter Conference on Applications of Computer Vision*, pp. 2665–2674, 2023.

Paul, S., Roy, S., and Roy-Chowdhury, A. K. W-talc: Weakly-supervised temporal activity localization and classification. In *Proceedings of the European Conference on Computer Vision*, pp. 563–579, 2018.

Pu, Y. and Wu, X. Locality-aware attention network with discriminative dynamics learning for weakly supervised anomaly detection. In *Proceedings of the IEEE International Conference on Multimedia and Expo*, pp. 1–6, 2022.

Pu, Y., Wu, X., Yang, L., and Wang, S. Learning prompt-enhanced context features for weakly-supervised video anomaly detection. *IEEE Transactions on Image Processing*, 33:4923–4936, 2024.

Radford, A., Kim, J. W., Hallacy, C., Ramesh, A., Goh, G., Agarwal, S., Sastry, G., Askell, A., Mishkin, P., Clark, J., et al. Learning transferable visual models from natural language supervision. In *Proceedings of the International Conference on Machine Learning*, pp. 8748–8763, 2021.

Rai, A. K., Krishna, T., Hu, F., Drimbarean, A., McGuinness, K., Smeaton, A. F., and O'connor, N. E. Video anomaly detection via spatio-temporal pseudo-anomaly generation: A unified approach. In *Proceedings of the IEEE Conference on Computer Vision and Pattern Recognition*, pp. 3887–3899, 2024.

Ramachandra, B., Jones, M. J., and Vatsavai, R. R. A survey of single-scene video anomaly detection. *IEEE Transactions on Pattern Analysis and Machine Intelligence*, 44 (5):2293–2312, 2020.

Sapkota, H. and Yu, Q. Bayesian nonparametric submodular video partition for robust anomaly detection. In *Proceedings of the IEEE Conference on Computer Vision and Pattern Recognition*, pp. 3212–3221, 2022.

Shou, Z., Gao, H., Zhang, L., Miyazawa, K., and Chang, S.-F. Autoloc: Weakly-supervised temporal action localization in untrimmed videos. In *Proceedings of the European Conference on Computer Vision*, pp. 154–171, 2018.

Sima, C., Renz, K., Chitta, K., Chen, L., Zhang, H., Xie, C., Beißwenger, J., Luo, P., Geiger, A., and Li, H. Drivelm: Driving with graph visual question answering. In *Proceedings of the European Conference on Computer Vision*, pp. 256–274, 2025.

Sultani, W., Chen, C., and Shah, M. Real-world anomaly detection in surveillance videos. In *Proceedings of the IEEE Conference on Computer Vision and Pattern Recognition*, pp. 6479–6488, 2018.

Tian, Y., Pang, G., Chen, Y., Singh, R., Verjans, J. W., and Carneiro, G. Weakly-supervised video anomaly detection with robust temporal feature magnitude learning. In *Proceedings of the IEEE International Conference on Computer Vision*, pp. 4975–4986, 2021.

Tran, D., Bourdev, L., Fergus, R., Torresani, L., and Paluri, M. Learning spatiotemporal features with 3d convolutional networks. In *Proceedings of the IEEE International Conference on Computer Vision*, pp. 4489–4497, 2015.

Van der Maaten, L. and Hinton, G. Visualizing data using t-sne. *Journal of Machine Learning Research*, 9(11), 2008.

Wang, J. and Cherian, A. Gods: Generalized one-class discriminative subspaces for anomaly detection. In *Proceedings of the IEEE International Conference on Computer Vision*, pp. 8201–8211, 2019.

Wang, Y., Li, Y., and Wang, H. Two-stream networks for weakly-supervised temporal action localization with semantic-aware mechanisms. In *Proceedings of the IEEE Conference on Computer Vision and Pattern Recognition*, pp. 18878–18887, 2023a.

Wang, Y., Zhao, S., and Chen, S. Action-semantic consistent knowledge for weakly-supervised action localization. *IEEE Transactions on Multimedia*, 26:10279–10289, 2024.

Wang, Y., Zhao, S., and Chen, S. Sql-net: Semantic query learning for point-supervised temporal action localization. *IEEE Transactions on Multimedia*, 27:84–94, 2025.

Wang, Z., Gao, Z., Guo, K., Yang, Y., Wang, X., and Shen, H. T. Multilateral semantic relations modeling for image text retrieval. In *Proceedings of the IEEE Conference on Computer Vision and Pattern Recognition*, pp. 2830–2839, 2023b.

Wei, Z., Pan, Z., and Owens, A. Efficient vision-language pre-training by cluster masking. In *Proceedings of the IEEE Conference on Computer Vision and Pattern Recognition*, pp. 26815–26825, 2024.

Wu, P. and Liu, J. Learning causal temporal relation and feature discrimination for anomaly detection. *IEEE Transactions on Image Processing*, 30:3513–3527, 2021.

Wu, P., Liu, J., Shi, Y., Sun, Y., Shao, F., Wu, Z., and Yang, Z. Not only look, but also listen: Learning multimodal violence detection under weak supervision. In *Proceedings of the European Conference on Computer Vision*, pp. 322–339, 2020.

Wu, P., Liu, X., and Liu, J. Weakly supervised audio-visual violence detection. *IEEE Transactions on Multimedia*, 25:1674–1685, 2022.

Wu, P., Liu, J., He, X., Peng, Y., Wang, P., and Zhang, Y. Toward video anomaly retrieval from video anomaly detection: New benchmarks and model. *IEEE Transactions on Image Processing*, 33:2213–2225, 2024a.

Wu, P., Zhou, X., Pang, G., Sun, Y., Liu, J., Wang, P., and Zhang, Y. Open-vocabulary video anomaly detection. In *Proceedings of the IEEE Conference on Computer Vision and Pattern Recognition*, pp. 18297–18307, 2024b.

Wu, P., Zhou, X., Pang, G., Yang, Z., Yan, Q., Wang, P., and Zhang, Y. Weakly supervised video anomaly detection and localization with spatio-temporal prompts. In *Proceedings of the ACM International Conference on Multimedia*, pp. 9301–9310, 2024c.

Wu, P., Zhou, X., Pang, G., Zhou, L., Yan, Q., Wang, P., and Zhang, Y. Vadclip: Adapting vision-language models for weakly supervised video anomaly detection. In *Proceedings of the AAAI Conference on Artificial Intelligence*, pp. 6074–6082, 2024d.

Wu, W., Wang, X., Luo, H., Wang, J., Yang, Y., and Ouyang, W. Bidirectional cross-modal knowledge exploration for video recognition with pre-trained vision-language models. In *Proceedings of the IEEE Conference on Computer Vision and Pattern Recognition*, pp. 6620–6630, 2023.

Xiao, J., Yao, A., Li, Y., and Chua, T.-S. Can i trust your answer? visually grounded video question answering. In *Proceedings of the IEEE Conference on Computer Vision and Pattern Recognition*, pp. 13204–13214, 2024.

Yang, Z., Liu, J., and Wu, P. Text prompt with normality guidance for weakly supervised video anomaly detection. In *Proceedings of the IEEE Conference on Computer Vision and Pattern Recognition*, pp. 18899–18908, 2024.

Yu, J., Liu, J., Cheng, Y., Feng, R., and Zhang, Y. Modality-aware contrastive instance learning with self-distillation for weakly-supervised audio-visual violence detection. In *Proceedings of the ACM International Conference on Multimedia*, pp. 6278–6287, 2022.

Zaheer, M. Z., Mahmood, A., Astrid, M., and Lee, S.-I. Claws: Clustering assisted weakly supervised learning with normalcy suppression for anomalous event detection. In *Proceedings of the European Conference on Computer Vision*, pp. 358–376, 2020.

Zaheer, M. Z., Mahmood, A., Khan, M. H., Segu, M., Yu, F., and Lee, S.-I. Generative cooperative learning for unsupervised video anomaly detection. In *Proceedings of the IEEE Conference on Computer Vision and Pattern Recognition*, pp. 14744–14754, 2022.

Zaheer, M. Z., Mahmood, A., Astrid, M., and Lee, S.-I. Clustering aided weakly supervised training to detect anomalous events in surveillance videos. *IEEE Transactions on Neural Networks and Learning Systems*, 35(10): 14085–14098, 2024.

Zhang, C., Li, G., Qi, Y., Wang, S., Qing, L., Huang, Q., and Yang, M.-H. Exploiting completeness and uncertainty of pseudo labels for weakly supervised video anomaly detection. In *Proceedings of the IEEE Conference on Computer Vision and Pattern Recognition*, pp. 16271–16280, 2023.

Zhang, J., Qing, L., and Miao, J. Temporal convolutional network with complementary inner bag loss for weakly supervised anomaly detection. In *Proceedings of the IEEE International Conference on Image Processing*, pp. 4030–4034, 2019.

Zhang, P., Li, X., Hu, X., Yang, J., Zhang, L., Wang, L., Choi, Y., and Gao, J. Vinvl: Revisiting visual representations in vision-language models. In *Proceedings of the IEEE Conference on Computer Vision and Pattern Recognition*, pp. 5579–5588, 2021.

Zhong, J.-X., Li, N., Kong, W., Liu, S., Li, T. H., and Li, G. Graph convolutional label noise cleaner: Train a plug-and-play action classifier for anomaly detection. In *Proceedings of the IEEE Conference on Computer Vision and Pattern Recognition*, pp. 1237–1246, 2019.

Zhong, Y., Zhu, R., Yan, G., Gan, P., Shen, X., and Zhu, D. Inter-clip feature similarity based weakly supervised video anomaly detection via multi-scale temporal mlp. *IEEE Transactions on Circuits and Systems for Video Technology*, 2024.

Zhou, H., Yu, J., and Yang, W. Dual memory units with uncertainty regulation for weakly supervised video anomaly

detection. In *Proceedings of the AAAI Conference on Artificial Intelligence*, pp. 3769–3777, 2023.

Zhu, J. and Pang, G. Toward generalist anomaly detection via in-context residual learning with few-shot sample prompts. In *Proceedings of the IEEE Conference on Computer Vision and Pattern Recognition*, pp. 17826–17836, 2024.

Zhu, L., Zheng, C., Guan, W., Li, J., Yang, Y., and Shen, H. T. Multi-modal hashing for efficient multimedia retrieval: A survey. *IEEE Transactions on Knowledge and Data Engineering*, 36(1):239–260, 2023.

