# OpenReview forum: "Learning Event Completeness for Weakly Supervised Video Anomaly Detection"
_ICML.cc/2025/Conference — ICML 2025 poster_

### Official Review · Reviewer_zcbm · 2025-03-05

**Overall Recommendation:** 3

**Summary:**

The main challenge in video anomaly detection is the lack of dense frame-level annotations, leading to incomplete localization in existing WS-VAD methods. To tackle this,  authors introduced Learning Event Completeness for WS-VAD, featuring a dual structure that captures both category-aware and category-agnostic semantics. It uses an anomaly-aware Gaussian mixture to define precise event boundaries and includes a memory bank-based mechanism to enhance text descriptions of anomaly-event categories. The proposed method shows promising results on the XD-Violence and UCF-Crime datasets.

## Update after rebuttal

With additional experiments, some of my concerns have been addressed. However, some of my questions remain, specifically regarding the loss function ablations. Therefore, I will keep my score as it is.

**Claims And Evidence:**

The proposed approach is somewhat incremental, and the results are promising. However, the problem formulation and motivation behind the approach are somewhat weak and need clearer articulation. The authors attempt to reformulate the problem in the second paragraph on page 1, but the proposed approach—which includes a dual structure  category-aware and category-agnostic semantics, an anomaly-aware Gaussian mixture to define precise event boundaries, and a memory bank-based mechanism to enhance text descriptions—requires further clarification with some of existing approaches. Additionally, comparing this approach with existing work is necessary to clearly address the challenges it aims to solve. This would provide a stronger foundation for the proposed approach and enhance its completeness.

The overall structure of the proposed approach presents some issues, as shown in Figure 2. Q represents textual features, while k and V represent video features, indicating that it involves cross-attention rather than self-attention in the Cross-Modal Aware block, which requires correction. On page 4, in the second column, the authors state "modeling cross-modal interactions through a cross-attention operation."

Furthermore, there is no clear formulation regarding the local transformer and global GCN. It remains unclear what the main contribution is beyond merely combining existing approaches to improve performance; the approach needs something novel to distinguish itself.

The experimental results support the contributions of the proposed approach, which outperforms existing methods by a large margin across different evaluation settings. The visualization results further demonstrate the advantages of the proposed approach.

**Essential References Not Discussed:**

The authors have included most of the related works in the experimental comparisons published in 2023 and 2024. The existing approaches are compared across different experimental settings. However, the authors need to discuss some of the related work in the introduction section to strengthen the argument and motivation for the proposed approach.

As I mentioned, the formulation of the problem and motivation is weak, which may explain why the authors did not include a discussion of the 2023 and 2024 papers.

**Experimental Designs Or Analyses:**

The experimental design is somewhat satisfactory, as the authors evaluate the approach in different settings, such as coarse-grained and fine-grained. However, there are some issues to address. For instance, in Table 1, only one unsupervised approach is presented. It is unclear whether the proposed approach is evaluated in an unsupervised manner. It appears that the evaluation is conducted with weakly supervised setting under different backbone networks, making the unsupervised row less relevant in this context.

In the fine-grained setting, it is essential to clarify the specific evaluation setting used for the proposed approach—whether it is weakly supervised, unsupervised, or something else. This needs to be explicitly stated.

Additionally, a major issue in the experimental settings is the evaluation of the proposed modules and the loss function formulation in Tables 5 and 6. These aspects need to be clearly presented for better understanding.

**Methods And Evaluation Criteria:**

The authors propose an approach to address the challenges in video anomaly detection by introducing Learning Event Completeness techniques. The proposed approach is evaluated in both coarse-grained and fine-grained settings, yielding very promising results, as shown in the quantitative and qualitative analyses. Furthermore, when compared to approaches from 2023 and 2024, the proposed method outperforms them by a large margin.

Some of the experimental settings require clarification. For example, in Table 5 ("Explorations of the Model Structure"), the authors evaluate the vision-only anomaly-aware branch and the cross-modal anomaly-aware branch. What are the differences between the last  rows and the second-to-last row in terms of experimental settings? If there are different settings, how did the authors evaluate them?

Furthermore, the main contribution of the paper is the dual structure that captures both category-aware and category-agnostic semantics. It is essential to clearly present how this approach is evaluated and to articulate the contributions of the proposed method more explicitly.

Additionally, in Table 6 ("Ablation Studies of Loss Function"), only two loss functions are presented. However, as defined in Equation 10, there are four loss functions. What are the contributions of the other loss functions, and why were they not evaluated? A clear explanation regarding this would be beneficial.

**Other Comments Or Suggestions:**

The authors need to carefully review the figures and table captions to ensure clarity for the reader. For example, in Figure 2, it should specify "cross-attention" rather than "self-attention," based on the features of Q, K, and V.

**Other Strengths And Weaknesses:**

The results of the proposed approach are significant; however, it appears to be a combination of existing methods. There is a lack of clear presentation in some sections, particularly in the proposed approach and the experimental settings.

For instance, the formulation of the local transformer and global GCN needs clarification. What are the main contributions of these modules? Additionally, it is important to explain how these components enhance the performance of the proposed approach and the overall contributions of the proposed modules. These aspects should be emphasized for better understanding.

**Questions For Authors:**

Most of my concerns are outlined in the above, and the authors need to address them. Specifically, in section three regarding the proposed approach and experimental settings, I have the following questions:

1.) The formulation of the local transformer and global GCN needs clarification. What are the main contributions of these modules? how it affect the proposed approach? It is important to explain how these components enhance the performance of the proposed approach and the overall contributions of the proposed modules.

2). In the loss function formulation, only two loss functions are evaluated and presented in the manuscript. What about the other loss functions?

3). What are the differences between the last rows and the second-to-last row in terms of experimental settings? If there are different settings, how did the authors evaluate them?

4) . As shown in Figure 2, there are three branches that are components of the proposed approach. However, in the ablation section, only VOB and CMB are evaluated. What about the other branch? What is the contribution of the proposed branch?

**Relation To Broader Scientific Literature:**

The authors attempt to address the main challenges of video anomaly detection with a dual structure that captures both category-aware and category-agnostic semantics. The proposed idea is intriguing, and the experimental results are promising, particularly in two main settings: coarse-grained and fine-grained.

However, a significant aspect of this paper is how the problem is formulated and the motivation behind it, where there appear to be gaps. As mentioned earlier, the novelty of the approach seems somewhat incremental. While the GMP-based Local Consistency Learning branch is a new contribution, its presentation is unclear, and there is a lack of strong emphasis on its significance.

**Theoretical Claims:**

The proposed approach makes sense but appears somewhat incremental, primarily consisting of a combination of existing methods. However, the results in both coarse-grained and fine-grained settings are interesting.

The approach lacks some theoretical explanations, particularly regarding the problem formulation and motivation for the dual structure that captures both category-aware and category-agnostic semantics. In Section 3, the vision branch includes a local transformer and global GCN, but this needs clearer presentation.

Additionally, it is important to explain how the main contributions enhance the learning of discriminative features. While some formulations are provided, they are not entirely clear. If these aspects were supported by a more robust theoretical explanation, it would significantly enhance the paper and clarify the concepts for the reader.

---

> ### Author Rebuttal · Authors · 2025-03-31
>
> Reviewer zcbm
>
> __Q1: The formulation of the local transformer and global GCN needs clarification. What are the main contributions of these modules? how does it affect the proposed approach?.__
>
> We appreciate the reviewer's insightful suggestions. The CLIP image encoder is primarily used to extract image features. But it exhibits limited proficiency in **modeling temporal dependencies inherent in videos**. To address this limitation, we integrate a local transformer layer alongside GCN modules to augment video features. We conduct ablation studies on GCN and Transformer modules and report the fine-grained AVG performance on XD-Violence:
>
> |GCN|Local Transformer|0.1|0.2|0.3|0.4|0.5|AVG|
> |-|:-:|:-:|:-:|:-:|:-:|:-:|:-:|
> |$\checkmark$||46.39|38.45|32.16|26.23|20.92|32.83|
> ||$\checkmark$|45.21|37.58|30.92|24.94|18.79|31.49|
> |$\checkmark$|$\checkmark$|**48.78**|**40.94**|**34.28**|**28.02**|**22.68**|**34.94**|
>
> In detail, we incorporate one local Transformer layer and 4 GCN layers. We present ablation studies that explore the rationale behind our choice and report the fine-grained AVG performance on XD-Violence:
> |Local Transformer/GCN|1|2|3|4|5|
> |:-:|:-:|:-:|:-:|:-:|:-:|
> |**1**|29.71|31.89|33.77|**34.94**|34.46|
> |**2**|28.24|30.27|32.03|33.15|32.76|
> |**3**|26.68|27.91|28.88|30.40|30.21|
>
> Also, we report fine-grained AVG results on UCF-Crime:
> | Local Transformer/GCN|1|2|3|4|5|
> |:-:|:-:|:-:|:-:|:-:|:-:|
> |**1**|10.46|11.67| 12.94|**13.56**|13.27|
> |**2**|9.37 |10.56|11.37|11.98|11.60|
> |**3**|8.01 |9.15|10.03|10.51|10.46|
>
> __Q2: In the loss function formulation, only two loss functions are evaluated and presented in the manuscript. What about the other loss functions?__
>
> Thanks for your comments. We do some ablation studies to evaluate the utilities of $L_{gmm}$ and $L_{reg}$ in Table 6. For the $L_{agnostic}$ and $L_{aware}$, since our model is designed to predict coarse-grained and fine-grained anomalies simultaneously, $L_{agnostic}$ and $L_{aware}$ are the basic terms used to train the model. Specifically, $L_{agnostic}$ is utilized to supervise coarse-grained anomaly detection, and $L_{aware}$ is utilized to supervise fine-grained anomaly detection. If either of them is removed, the model cannot detect anomaly events of the corresponding granularity (coarse-grained and fine-grained).
>
> **Q3: In Table 5, what are the differences between the last rows and the second-to-last row in terms of experimental settings? If there are different settings, how did the authors evaluate them?**
>
> Thanks for your comments. The last row of Table 5 indicates that the proposed algorithm leverages both visual information and textual descriptions of anomaly categories for both coarse-grained and fine-grained detections, while the second-to-last row only uses the visual information of the video. There is no difference in the experimental settings of other parameters.
>
> __Q4: As shown in Figure 2, there are three branches that are components of the proposed approach. However, in the ablation section, only VOB and CMB are evaluated. What about the other branch?__
>
> Thanks for your suggestions. In the ablation section, we evaluated the VOB and CMB, and the other branch is the proposed Gaussian Mixture Prior-based Local Consistency Learning mechanism, which is only used to guide the model's training. We explore ablation studies about this module by controlling the loss term $\mathcal{L}_{gmm}$ in Table 6. For clarity, we add several experiments and put them in a table for comparisons:
>
> |GMP|VOB|CMB|0.1|0.2|0.3|0.4|0.5|AVG|
> |-|:-:|:-:|:-:|:-:|:-:|:-:|:-:|:-:|
> ||$\checkmark$||37.23|30.96|23.77|18.03|14.33|24.86|
> |||$\checkmark$|38.61|31.54|24.04|18.67|14.81|25.53|
> |$\checkmark$|$\checkmark$||42.86|33.82|30.58|24.66|18.92|30.17|
> |$\checkmark$||$\checkmark$|46.03|38.87|32.18|27.02|22.04|33.23|
> ||$\checkmark$|$\checkmark$|46.69|39.51|32.17|27.16|22.01|33.51|
> |$\checkmark$|$\checkmark$|$\checkmark$|**48.78**|**40.94**|**34.28**|**28.02**|**22.68**|**34.94**|
>
> __Q5: The authors need to discuss some of the related work in the introduction section to strengthen the motivation for the proposed approach.__
>
> We appreciate the reviewer's insightful suggestions. **We will strengthen the discussion about our motivation in the camera-ready version**. Here, we also re-emphasize it. We observe the existing methods predict incomplete and fragmented event segments, and an example is also provided in Figure 1. The core reason for this phenomenon is the lack of frame-wise annotations in the weakly supervised learning framework, resulting in sparse semantics. For this reason, we propose a novel Gaussian mixture prior-based strategy and memory bank-based prototype mechanism to mitigate this issue.
>
> __Q6:The authors need to carefully review the figures and table captions to ensure clarity for the reader".__
>
> Thanks for your suggestions. We will modify these typos and inaccurate expressions.

---

### Official Review · Reviewer_MBS1 · 2025-03-10

**Overall Recommendation:** 3

**Summary:**

This paper is dedicated to improving anomaly detection performance by enhancing the completeness of predicted events. A dual-branch structure is introduced to capture both category-aware and category-agnostic semantics between vision and language. A prototype learning mechanism based on a memory bank is proposed to improve the representation of textual features. Besides, the approach utilizes learnable Gaussian masks to achieve local consistency in predictions.

**Claims And Evidence:**

The performance of the proposed method in coarse-grained and fine-grained anomaly detection has been effectively validated on two datasets. However, the mechanism of the introduced dual structure and memory bank-based prototype learning for improving the completeness of event predictions have not been clearly stated. In addition, in Figure 5, more examples are needed to demonstrate the improvement in the completeness of event predictions.

**Essential References Not Discussed:**

Please refer to the comments on 'Relation to Broad Scientific Literature'.

**Experimental Designs Or Analyses:**

The performance of the proposed method in coarse-grained and fine-grained anomaly detection has been effectively validated on two datasets. In addition, the ablation experiments utilize the setting where only one module is ablated independently, which is not ideal for verifying the relationships between each contribution. It is recommended to provide ablation experiments that incrementally introduce each module. Moreover, the visualizations in Figure 3 could be enhanced by including additional details, such as the source of the visualized features, particularly indicating whether the visualized normal features include those from normal segments in anomalous videos. Meanwhile, this visualization demonstrates the discriminative visual representations for different anomaly categories. However, in the event completeness visualization presented in Figure 5, which is a binary classification result, the analysis lacks the explanation of how the representations for different anomaly categories contribute to enhancing event completeness.

**Methods And Evaluation Criteria:**

Based on the provided main results and visual experiments, the proposed method appears to be meaningful in improving the performance of WSVAD. Besides, since the dual structure has already been introduced in VadCLIP, it would not be appropriate to present it as a primary contribution.

**Other Comments Or Suggestions:**

Memory bank-based prototype learning and GMP-based local consistency learning are introduced alternately in Section 3, though the structure of this section could be improved for better clarity and flow. Additionally, the tense usage should be checked, as both present and past tenses are used inconsistently in some paragraphs, such as in the conclusion.

**Other Strengths And Weaknesses:**

Strengths:
To the best of our knowledge, this paper is the first to introduce the completeness of events in predictions, which has been the issue also emphasized in other temporal localization tasks, to WSVAD. The issue addressed has indeed limited the performance of WSVAD but has been overlooked. Based on the visualization experiments, the proposed method learns discriminative features and achieves more accurate and complete predictions.

Weaknesses:
More comprehensive ablation studies should be conducted to further validate the effectiveness of the proposed modules. The mechanism of the dual structure and memory bank-based prototype learning in enhancing the completeness of event predictions is expected to be clearly explained and validated.

**Questions For Authors:**

Q1: The dual structure has already been proposed in VadCLIP, so it would be preferable not to present it as a primary contribution.
Q2: The ablation experiments use the setting where only one module is ablated independently, which is not ideal for verifying the relationships between each contribution. It is recommended to provide ablation experiments that incrementally introduce each module.
Q3: Multiple dense anomalous events may exist in an anomalous video, such as some testing videos in XD-Violence, and it is important to evaluate whether the proposed method might incorrectly classify these frequent anomalies as a single anomalous event.
Q4: The visualizations in Figure 3 should provide more details, such as the source of the visualized features, particularly clarifying whether the visualized normal features include those from normal segments within anomalous videos. Meanwhile, this visualization demonstrates the discriminative visual representations for different anomaly categories. However, in the event completeness visualization presented in Figure 5, which is a binary classification result, the analysis lacks the explanation of how the visual representations for different anomaly categories contribute to enhancing event completeness.

**Relation To Broader Scientific Literature:**

The dual-branch structure has already been proposed in VadCLIP. The proposed method should emphasize the difference between its dual-branch framework and that in VadCLIP, with experiments highlighting the difference contribute to performance improvements. Additionally, as the issue of event completeness is extensively studied in the Weakly Supervised Temporal Action Localization task, citing these works can convincingly introduce event completeness into WSVAD.

**Theoretical Claims:**

No

---

> ### Author Rebuttal · Authors · 2025-03-31
>
> Reviewer MBS1
>
> __Q1: The dual structure has already been proposed in VadCLIP, so it would be preferable not to present it as a primary contribution.__
>
> Thanks for your suggestions. We will update it on the camera-ready version.
>
> __Q2: The ablation experiments use the setting where only one module is ablated independently, which is not ideal for verifying the relationships between each contribution. It is recommended to provide ablation experiments that incrementally introduce each module.__
>
> We appreciate the reviewer's insightful suggestion. Here, we provide an extra ablation study on XD-Violence to verify the relationships between each contribution. We have observed that the proposed modules can bring positive gains.
>
> |VOB|CMB|VAP|PMB|0.1|0.2|0.3|0.4|0.5|AVG|
> |-|:-:|:-:|:-:|:-:|:-:|:-:|:-:|:-:|:-:|
> |$\checkmark$||||33.74|28.57|22.45|17.81|13.40|23.19|
> ||$\checkmark$|||39.23|32.96|25.77|20.03|16.33|26.86|
> |$\checkmark$|$\checkmark$|||42.17|34.66|27.98|22.47|17.24|28.90|
> |$\checkmark$|$\checkmark$|$\checkmark$||46.24|38.75|31.97|26.63|20.05|32.73|
> |$\checkmark$|$\checkmark$||$\checkmark$|47.67|39.98|32.85|27.59|21.00|33.82|
> |$\checkmark$|$\checkmark$|$\checkmark$|$\checkmark$|**48.78**|**40.94**|**34.28**|**28.02**|**22.68**|**34.94**|
>
>
> __Q3: Multiple dense anomalous events may exist in an anomalous video, such as some testing videos in XD-Violence, and it is important to evaluate whether the proposed method might incorrectly classify these frequent anomalies as a single anomalous event.__
>
> We appreciate the reviewer's insightful suggestion. For fine-grained anomaly detection, the model is asked to predict anomalous events about each category, as reported in Table 3 and Table 4. Here, we divide the original test set (800 videos) into two parts. One part contains videos with only one type of abnormal events, and the other part contains videos with multiple types of abnormal events. The former contains 753 videos, while the latter contains 47 videos and is more challenging. We report fine-grained prediction results in these 47 videos that contain multiple anomaly categories:
>
> |Methods|0.1|0.2|0.3|0.4|0.5|AVG|
> |-|:-:|:-:|:-:|:-:|:-:|:-:|
> |VadCLIP|35.37|26.36|23.85|6.03|4.46|19.21|
> |ReFLIP|39.38|26.96|24.73|14.47|9.89|23.09|
> |Ours|**41.68**|**28.32**|**27.47**|**20.20**|**15.37**|**26.61**|
>
> We observe that our proposed method achieves significant improvement, especially when the the evaluation criteria are more stringent, namely larger tIoU values.
>
>  __Q4: The visualizations in Figure 3 should provide more details, such as the source of the visualized features, particularly clarifying whether the visualized normal features include those from normal segments within anomalous videos. Meanwhile, this visualization demonstrates the discriminative visual representations for different anomaly categories. However, in the event completeness visualization presented in Figure 5, which is a binary classification result, the analysis lacks the explanation of how the visual representations for different anomaly categories contribute to enhancing event completeness.__
>
>  We sincerely appreciate the reviewer's insightful comments and constructive suggestions regarding the visualizations in our paper. Below, we address the specific concerns raised about Figure 3 and Figure 5.
>
> For Figure 3, we acknowledge the reviewer's request for clarification on the source of the visualized features. The visualized features are the enhanced feature $X_{l}$ learned by our model, and the source of them are features extracted by the CLIP image encoder. Besides, the visualized normal features include those from normal segments within anomalous videos. As shown in Figure 3, these features form clusters well, further proving that our learned features are highly discriminative. **We will revise the caption of Figure 3 for these suggestions in the camera-ready version**.
>
> Although Figure 5 highlights binary classification results, the model utilizes the distinctive representations learned for different anomaly categories (depicted in Figure 3) to inform its decisions. This capacity to differentiate among anomaly types indirectly helps delineate the boundaries and attributes of anomalous events.
>
> Furthermore, **In the anonymous link https://anonymous.4open.science/r/Visualization_ICML_Rebuttal-B61B/README.md, we also provide some visualization examples to analyze the influence of different anomaly categories on enhancing event completeness.**
>
>
>  __Q5: Some other suggestions about expressions and the section structure.__
>
>  Thanks for your valuable suggestion. We will improve the structure of Section 3 and some expressions in this paper in the revised version.

---

### Official Review · Reviewer_K2pV · 2025-03-11

**Overall Recommendation:** 3

**Summary:**

This paper proposes a new WSVAD framework, LEC-VAD, that utilizes visual and language modalities for category-agnostic and category-aware anomaly detection. The authors employ a Gaussian mixture method to guide the model in predicting more complete anomaly boundaries. Additionally, a memory bank-based prototype learning mechanism is introduced to enhance the text feature representation related to anomalies. LEC-VAD achieves state-of-the-art performance in both coarse-grained and fine-grained results on the XD-Violence and UCF-Crime datasets.

**Claims And Evidence:**

The claims are mostly clear and convincing.

**Essential References Not Discussed:**

The cited baselines are comprehensive and include recent WS-VAD methods.

**Experimental Designs Or Analyses:**

The paper includes comprehensive experiments: comparisons across feature backbones, ablation studies, and hyperparameter analysis. However, all ablation studies focus solely on fine-grained detection results, lacking ablation experiments on coarse-grained performance.

**Methods And Evaluation Criteria:**

The approach is conceptually sound, and the evaluation datasets align with community standards.

**Other Comments Or Suggestions:**

None.

**Other Strengths And Weaknesses:**

Strengths:

1.The paper is well-structured and clearly written.

2.The motivation—detecting more precise anomaly boundaries—is reasonable and addresses a critical limitation in WS-VAD.

3.The proposed method achieves state-of-the-art performance across multiple benchmarks and evaluation metrics.

Weaknesses:

1.Problem setting: The fine-grained detection assumes closed-set anomaly categories are available, this is a common practice in the Temporal Action Localization (TAL) field.  However, the reliance on category labels in Video Anomaly Detection (VAD) domain may limits the generalizability of the trained model, since most abnormal categories are unknown/unpredictable in real-world application (the key difference between VAD and TAL ).

2.Motivation: While the title and motivation emphasize "Event Completeness," the Gaussian mixture mechanism primarily enforces local consistency rather than explicitly addressing boundary completeness. Visualizations of Gaussian-rendered anomaly score and its impact on boundary precision would strengthen this claim.

3.Method：The proposed Guassian mixture and memory-based prototype techniques has been used in other video action/anomaly detection methods[1-4]. Their adaptation to VAD lacks domain-specific innovations or insights compared to prior work in action detection.

4.Experiments: As noted earlier, the absence of coarse-grained ablation studies weakens the evaluation.

[1]Gaussian temporal awareness networks for action localization. CVPR2019.
[2]GlanceVAD: Exploring glance supervision for label-efficient video anomaly detection.
[3]HR-Pro: Point-supervised temporal action localization via hierarchical reliability propagation. AAAI2024.
[4]Anomaly Detection with Prototype-Guided Discriminative Latent Embeddings.

Overall, the contributions in the current version of paper are limited. However, if the authors address my concerns, I may consider increasing my rating.

**Questions For Authors:**

The XD-Violence dataset contains videos with multiple anomaly categories and events, and labels are not available for each anomaly event. Could the authors clarify how mAP is computed in this multi-label scenario?

**Relation To Broader Scientific Literature:**

The key contributions include the Gaussian mixture prior for smoother local supervision and the memory-based prototype mechanism for feature enrichment. However, similar techniques (e.g., Gaussian kernels for temporal modeling [1] and smoother guidance[2], memory-based prototypes for representation enhancement [3,4]) have been explored in action/anomaly detection literature. The authors need to clarify how these adaptations differ in the context of anomaly detection or provide novel insights specific to VAD.

[1]Gaussian temporal awareness networks for action localization. CVPR2019.
[2]GlanceVAD: Exploring glance supervision for label-efficient video anomaly detection.
[3]HR-Pro: Point-supervised temporal action localization via hierarchical reliability propagation. AAAI2024.
[4]Anomaly Detection with Prototype-Guided Discriminative Latent Embeddings.

**Theoretical Claims:**

N/A.

---

> ### Author Rebuttal · Authors · 2025-03-31
>
> __Q1: The fine-grained detection assumes closed-set categories. However, the reliance on category labels in VAD may limit the generalizability, since most abnormal categories are unknown/unpredictable.__
>
> Thanks for your suggestion. We agree that an over-reliance on category labels could potentially limit the generalizability. However, our proposed two mechanisms can mitigate the risks, including the usage of the  CLIP text encoder and the prototype-based memory bank mechanism (PMB). First, CLIP itself can recognize open-set categories. Second, PMB can be understood to some extent as identifying general patterns and characteristics of anomalies, instead of focusing on learning specific categories.
>
> To verify the generalization, we report the AUC on **an extra dataset UBnormal**, where only 7 abnormal categories are visible during training and 12 abnormal categories are used for testing. The results show our model can efficiently handle open-set anomaly categories, especially when using PMB.
> |Methods|AUC(%)|
> |-|:-:|
> |RAD|50.30|
> |Wu et al.[1]|53.70|
> |DMU|59.91|
> |RTFML|60.94|
> |ReFLIP|61.13|
> |**Ours w/o PMB**|60.85|
> |**Ours w/ PMB**|**61.65**|
>
> [1] Not only look, but also listen: Learning multimodal violence detection under weak supervision. ECCV
>
> __Q2: The Gaussian mixture mechanism primarily enforces local consistency rather than explicitly addressing completeness. Visualizations of Gaussian-rendered scores would strengthen this claim.__
>
> Thanks for your suggestions. We define the average tIoU between predictions and GT for all classes' instances as a metric of the Event Completeness (EC) and report fine-grained results on XD-Violence:
> |Methods|EC(%)|
> |-|:-:|
> |Ours w/o GMP|12.6|
> |Ours w/ GMP|**15.5**|
>
> We offer visual comparisons that illustrate the differences in outcomes when incorporating Gaussian-based scores versus not using them in https://anonymous.4open.science/r/icml_vis-CDD6/README.md
>
> Besides, we also provide visualizations for various categories in https://anonymous.4open.science/r/Visualization_ICML_Rebuttal-B61B/README.md We find that our model can get desirable results.
>
> __Q3: The proposed Gaussian mixture and memory-based prototype techniques.__
>
> We appreciate the reviewer's insightful observations. First, both [1] and [2] are used in VAD instead of WS-VAD. Besides, they only model a **unimodal Gaussian model** with complex kernel function designs, but our method models prediction scores of multiple anomaly categories as **Gaussian Mixture Model** to ensure local consistency of predictions. With GMM, our model can learn the correlations between multiple anomaly categories, but the unimodal Gaussian model cannot.
>
> For the memory-based prototype technique, [3] uses point annotations of action clips to create a memory bank storing reliable visual prototypes with threshold strategies. However, our weakly supervised scenario cannot access point annotations, so our memory-based is designed to build flexible text representations of anomaly categories. This can be also understood as general patterns of learning abnormal behaviors, as we explained in Q1. Besides, [4] learned prototype-guided discriminative embeddings to separate normal and abnormal data. It uses a different paradigm and purpose, which is essentially different from our method.
>
> __Q4: The absence of coarse-grained ablation studies weakens the evaluation.__
>
> Thanks for your suggestions. Due to the page limit, we showed more challenging fine-grained results in the original paper. Here, we provide coarse-grained results using I3D features on XD-Violence
>
> [1] Ablation studies of the model structure:
> |VOB|COM|AP(%)|
> |-|:-:|:-:|
> |✅||84.85|
> ||✅|87.10|
> |✅|✅|**88.47**|
>
> [2] Ablation studies of the loss function:
> |$L_{reg}$|$L_{gmm}$|AP(%)|
> |-|:-:|:-:|
> |||85.43|
> |✅||86.95|
> ||✅|87.79|
> |✅|✅|**88.47**|
>
> [3] Ablation studies of text enhancement:
> |VAP|PMB|AP(%)|
> |-|:-:|:-:|
> |||82.14|
> |✅||86.69|
> ||✅|86.83|
> |✅|✅|**88.47**|
>
> We observe that coarse-grained results show consistent conclusions with fine-grained versions, which further reveals the utilities of the proposed components.
>
> __Q5: Multi-label evaluation for XD-Violence dataset.__
>
> We use the same evaluation paradigm and codes as prior works like VadCLIP, ReCLIP, etc. Among 800 test videos, 47 samples have multiple labels. As labels for individual anomaly events aren't provided, the current evaluation paradigm only computes mAP without considering classes, which may introduce bias. However, since the proportion of such multi-label cases is low and all current methods follow this paradigm, the comparison remains fair. We also evaluated results only on 47 videos and found that its results were far lower than those on 800 videos, which reveals its results did not play a leading role.
>
> |Methods|0.1|0.2|0.3|0.4|0.5|AVG|
> |-|:-:|:-:|:-:|:-:|:-:|:-:|
> |VadCLIP|35.37|26.36|23.85|6.03|4.46|19.21|
> |ReFLIP|39.38|26.96|24.73|14.47|9.89|23.09|
> |Ours|**41.68**|**28.32**|**27.47**|**20.20**|**15.37**|**26.61**|

---

### Official Review · Reviewer_QCZn · 2025-03-12

**Overall Recommendation:** 3

**Summary:**

This paper explores weakly supervised video anomaly detection, introducing a dual-structure framework that captures both category-aware and category-agnostic semantics through vision-language integration. To enhance anomaly scoring, the authors propose a learnable Gaussian mixture mask that produces smoother scoring patterns. The effectiveness of the proposed approach is validated on standard benchmark datasets, including UCF-Crime and XD-Violence.

**Claims And Evidence:**

The experimental evaluation and ablation study are thorough, effectively verifying the contribution of each component and the influence of hyperparameters in the objective function.

However, the paper could be further improved by explicitly demonstrating the impact of the local transformer layer and the GCN module in enhancing video features, which is currently not addressed. Additionally, the rationale behind the choice of 𝐾 in top-𝐾 scores and their impact remain unclear to the reviewer.

**Essential References Not Discussed:**

Please consider including recent publications in this field, particularly those focused on LLM- or VLM-based approaches, along with relevant discussions.

**Experimental Designs Or Analyses:**

The C3D result for LEC-VAD is missing in Table 1. Please clarify the reason.

**Methods And Evaluation Criteria:**

The inference time of the proposed method is not provided.

Additionally, the definitions of the vision-only anomaly-aware branch (VOB) and the cross-modal anomaly-aware branch (CMB) are unclear. Are the authors referring to VOB as the component that outputs F_{video}, or does it serve a different purpose? Please clarify.

**Other Comments Or Suggestions:**

Please provide a detailed comparison between the proposed method and ReFLIP-VAD.

Additionally, consider enlarging the text in Figure 3 for improved readability.

**Other Strengths And Weaknesses:**

Please clarify the concept of Event Completeness in the proposed approach, as it remains unclear. At the very least, the authors should justify how their method addresses this aspect.

**Questions For Authors:**

Please refer to the above comments.

**Relation To Broader Scientific Literature:**

This study could be valuable to the video anomaly detection community. Additionally, some of the paper's empirical design choices for post-processing may also benefit the action detection community.

**Theoretical Claims:**

This paper primarily focuses on the empirical design of improved architectural structures and learning objectives, without making any theoretical claims.

However, Equation 8 could be further justified, particularly regarding the choice to regularize the predicted anomaly scores rather than the ground truth scores, as the latter may appear to be a more suitable option.

---

> ### Author Rebuttal · Authors · 2025-03-31
>
> Reviewer QCZn
>
> __Q1: The impact of the local transformer layer and the GCN module in enhancing video features.__
>
> Thanks for your advice. The CLIP image encoder is primarily used to extract image features. However, it exhibits limited proficiency in modeling temporal dependencies inherent in videos. To address this limitation, we integrate a local transformer layer alongside GCN modules to augment video features. In detail, we incorporate one local Transformer layer and 4 GCN layers. In the following table, we present ablation studies that explore the rationale behind our choice and report the fine-grained AVG performance on XD-Violence.
> |Local Transformer/GCN|1|2|3|4|5|
> |:-:|:-:|:-:|:-:|:-:|:-:|
> |**1**|29.71|31.89|33.77|**34.94**|34.46|
> |**2**|28.24|30.27|32.03|33.15|32.76|
> |**3**|26.68|27.91|28.88|30.40|30.21|
>
> Also, we report fine-grained AVG results on UCF-Crime:
> |Local Transformer/GCN|1|2|3|4|5|
> |:-:|:-:|:-:|:-:|:-:|:-:|
> |**1**|10.46|11.67| 12.94|**13.56**|13.27|
> |**2**|9.37 |10.56|11.37|11.98|11.60|
> |**3**|8.01 |9.15|10.03|10.51|10.46|
>
> __Q2: The rationale behind the choice of 𝐾 in top-𝐾 scores__
>
> In this paper, we set $K=max(\lfloor T/16\rfloor$), as explained in Implementation Details. This decision stems from our utilization of multi-instance learning (MIL) to get video-level outcomes. Previous works that used MIL used the same setup [1,2], which can **adapt to the length of videos**. The denominator uses 16 because these two datasets sample 16 consecutive frames into one segment during feature preprocessing.
>
> [1] Two-Stream Networks for Weakly-Supervised Temporal Action Localization with Semantic-Aware Mechanisms, CVPR 2023
> [2] Vadclip: Adapting vision-language models for weakly supervised video anomaly detection, AAAI 2024
>
> __Q3: The inference time of the proposed method__
>
> Thanks for your advice. I provide the average inference time (in seconds) of each video for fine-grained predictions on XD-Violence and UCF-Crime, and compare results with representative methods with a single Nvidia V100 GPU:
> |Methods|Inference Time on XD-Violence|Inference Time on UCF-Crime|
> |-|:-:|:-:|
> |VadCLIP|0.21|0.43|
> |ReFLIP|0.38|0.63|
> |ours|0.23|0.46|
>
> We observe that our method outperforms the SOTA ReFLIP in terms of speed. When compared to VadCLIP, it incurs only a marginal increase in processing time, yet achieves substantial improvements in performance.
>
> **Q4: The definitions of VOB and the CMB.**
>
> Thanks for your comments. VOB denotes both coarse-grained and fine-grained predictions derived exclusively from visual information, without incorporating category text descriptions. Instead, CMB combines visual inputs and text descriptions of anomaly classes.
>
> __Q5: Eq. 8 is chosen to regularize the predicted scores rather than ground truth.__
>
> Thanks for your comments. Eq.8 acts as a regularizer on the prediction scores and should not be used for GT. We hope to constrain the consistency of coarse-grained and fine-grained predictions by using Equation 8
>
> __Q6: The C3D result for LEC-VAD is missing in Table 1.__
>
> Thanks for your advice. We apologize for the omission of this result in the original paper. To rectify this, we will include the C3D results for our method in Table 1. This result is also presented here:
> |Modality|Methods|Features| AP(\%)|
> |:-:|:-:|:-:|:-:|
> |RGB|LEC-VAD|C3D|79.58|
>
> __Q7: Discussions about publications including LLM- or VLM-based approaches.__
>
> Thanks for your advice. Some LLM- or VLM-based methods are used to generate abundant and diverse anomaly class descriptions. Notably, these works introduce **additional data and knowledge** provided by LLM, and the results based on them may be unfair comparisons. This introduction of external expert knowledge is not in the scope of this article. However, **we will add relevant works and discussions in the related work**.
>
> __Q8: The concept of Event Completeness and how our method addresses this aspect.__
>
> Event completeness can be conceptualized as the proportion of predicted intervals relative to the total actual event intervals. It can be reflected when using larger tIoU values. As shown in Table 3 and Table 4, compared with ReFlip, our method achieves more **relative improvements** for larger tIoU values.
>
> __Q9: A detailed comparison between the proposed method and ReFLIP-VAD.__
>
> The core idea of ReFLIP-VAD includes: 1) modeling the global and local temporal dependencies; and 2) designing learnable prompt templates to provide interpretable and informative class clues. However, our method focuses on **the event completeness**. While our focus is also on generating informative text descriptions, we introduce an innovative memory bank-based mechanism, eschewing the use of prompt templates generated by an additional pre-trained encoder.
>
> __Q10: Consider enlarging the text in Figure 3 for improved readability.__
>
> Thanks for your advice. We will improve them in the camera-ready version.

---

### Decision · Program_Chairs · 2025-05-01

**Decision:**

Accept (poster)

**Comment:**

All four reviewers recommended weak accept. The reviewers generally appreciated the proposed idea and the experimental results, particularly in two main settings: coarse-grained and fine-grained.The reviewers also raised several questions in the initial review including,  ablation experiments on coarse-grained setting, multi-label evaluation results, ablations when incrementally introducing each module, and questions related to clarity in the presentation. Post-rebuttal, reviewers mentioned that most of their concerns are addressed. Given the manuscript, the rebuttal, and that all reviewers are generally positive, the AC agrees with the reviewers and recommend acceptance. Authors are encouraged to take into account all the suggestions of reviewers and rebuttal when preparing the revised manuscript.